# When Priors Backfire: On the Vulnerability of Unlearnable Examples to Pretraining

**Zhihao Li**[1] **Gezheng Xu**[1,*] **Jiale Cai**[1] **Ruiyi Fang**[1] **Di Wu**[2] **Qicheng Lao**[3]
**Charles Ling**[1,4] **Boyu Wang**[1,4]
[1] Western University     [2] Concordia University
[3] Beijing University of Posts and Telecommunications [4] Vector Institute
{zli3446,gxu86,jcai336,rfang32,charles.ling}@uwo.ca
di.wu@concordia.ca   qicheng.lao@bupt.edu.cn   bwang@csd.uwo.ca

## ABSTRACT

Unlearnable Examples (UEs) serve as a data protection strategy that generates imperceptible perturbations to mislead models into learning spurious correlations instead of underlying semantics. In this paper, we uncover a fundamental vulnerability of UEs that emerges when learning starts from a pretrained model. Crucially, our empirical analysis shows that even when data are protected by carefully crafted perturbations, pretraining priors still furnish rich semantic representations that allow the model to circumvent the shortcuts introduced by UEs and capture genuine features, thereby nullifying unlearnability. To address this, we propose **BAIT** (**B**inding **A**rtificial perturbations to **I**ncorrect **T**argets), a novel bi-level optimization formulation. Specifically, the inner level aims at associating the perturbed samples with real labels to simulate standard data-label alignment, while the outer level actively disrupts this alignment by enforcing a mislabel-perturbation binding that maps samples to designated incorrect targets. This mechanism effectively overrides the semantic guidance of priors, forcing the model to rely on the injected perturbations and consequently preventing the acquisition of true semantics. Extensive experiments on standard benchmarks and multiple pretrained backbones demonstrate that BAIT effectively mitigates the influence of pretraining priors and maintains data unlearnability. Code is available at https://github.com/zhli-cs/BAIT.

## 1 INTRODUCTION

The rapid growth of social media has produced massive online data, enabling large-scale datasets (Deng et al., 2009; Lin et al., 2014) and propelling data-driven deep learning (Ye et al., 2024; Xu et al., 2025b; Ji et al., 2025). However, it also raises concerns over unauthorized usage. To safeguard privacy, Unlearnable Examples (UEs) have emerged as a promising data protection strategy (Huang et al., 2021; Yu et al., 2025). By injecting perturbations into training data, UEs induce models to capture spurious shortcuts rather than underlying semantics, thereby degrading clean test performance.

Existing studies (Yu et al., 2022; Liu et al., 2024a) primarily target randomly initialized models. However, as training from scratch is annotation-intensive, practitioners commonly adopt pretrained backbones (Iofinova et al., 2022; Fang et al., 2023; Lu et al., 2024). Yet the behavior of UEs on pretrained models remains unexplored, leaving a critical gap in assessing their real-world applicability. Meanwhile, UEs essentially operate by injecting spurious shortcuts (Huang et al., 2021; Ren et al., 2023), and recent studies indicate that pretraining can improve robustness to spurious correlations (Tu et al., 2020; Izmailov et al., 2022; Mehta et al., 2022). This motivates us to raise a natural question: **Can unlearnable examples remain effective when applied to pretrained models?**

In this paper, we uncover an overlooked yet fundamental vulnerability of UEs when applied to a pretrained model. As shown in Figure 1a, pretrained models exhibit substantially higher test accuracy compared with their train-from-scratch counterparts, indicating that they circumvent the unlearnability and acquire authentic semantics. We hypothesize that pretraining priors are the primary factor

---
*Corresponding author

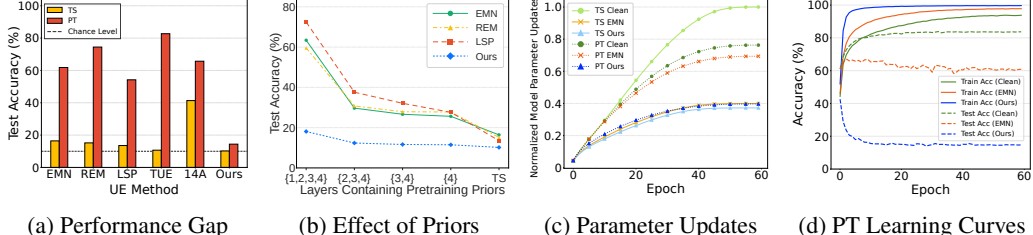

| (a) Performance Gap | (b) Effect of Priors | (c) Parameter Updates | (d) PT Learning Curves |

Figure 1: Empirical analysis of the vulnerability of UEs to pretraining priors. All experiments are conducted on CIFAR-10 with ResNet-18. (a) Existing UE methods suffer severe unlearnability degradation when applied to pretrained (PT) backbones instead of train-from-scratch (TS) models. (b) We progressively replace layers of a four-layer ImageNet pretrained ResNet-18 with randomly initialized layers until obtaining a train-from-scratch model. The resistance to UEs steadily diminishes as pretraining priors are removed. (c) We report the normalized model parameter updates when training on clean data and UEs (details in Appendix B.1). We observe that effective perturbations result in minimal parameter updates. In contrast, pretrained models bypass the spurious correlations induced by EMN (Huang et al., 2021) and remain fully optimized. (d) We plot the learning curve starting from a pretrained backbone. The concurrent rise in the training and test accuracy of EMN indicates that such substantial parameter updates drive the model to acquire real semantics rather than the injected shortcuts, thereby nullifying the unlearnability.

that enables models to bypass the UE-induced spurious correlations. To investigate their roles, we progressively remove priors by replacing the pretrained layers with randomly initialized ones and observe that the test accuracy consistently decreases, indicating that the unlearnability strengthens as priors are reduced, as illustrated in Figure 1b. We further posit that the presence of pretraining priors facilitates a semantic-label pathway, which guides models to exploit underlying semantics rather than the perturbation-label shortcuts, thereby enabling the resumption of effective learning. To validate this, we examine the parameter update dynamics when training on clean data versus UEs generated by EMN (Huang et al., 2021), as shown in Figure 1c. Our observations reveal that for pretrained models, the parameter update magnitude on UEs parallels that of clean data and considerably exceeds that of train-from-scratch counterparts. To delve deeper, we analyze the learning curve of a pretrained model in Figure 1d and observe that EMN demonstrates prominently higher test accuracy. This implies that such substantial parameter updates propel the model to learn real representations. Consequently, driven by the semantic pathway inherent in priors, models are able to resume learning meaningful image features rather than being confined to the UE-introduced spurious correlations. These findings highlight that existing UE methods are vulnerable to pretraining, as strong priors enable models to bypass their protection and acquire real semantics.

To address the aforementioned limitation, we propose **BAIT** (**B**inding **A**rtificial perturbations to **I**ncorrect **T**argets), a novel bi-level optimization framework designed to counteract pretraining priors. The core idea is to disrupt the standard data-label alignment reinforced by priors and instead recover the UE-induced spurious perturbation-label correlation. To achieve this, we devise a mislabel-perturbation binding mechanism that explicitly associates perturbations with incorrect labels that are semantically distinct from the ground truth. Specifically, the inner level utilizes pretraining priors to simulate standard data-label correspondences, while the outer level optimizes perturbations to map perturbed images onto designated incorrect targets, overriding the underlying semantic-label dependency induced by pretraining priors. This mechanism blocks models from readily exploiting pretraining priors, compelling reliance on perturbations and impeding the acquisition of meaningful semantics. As demonstrated in Figure 1, our method successfully reduces the test accuracy to chance level in the presence of pretraining priors. Our main contributions are summarized as follows:

- We reveal that UEs suffer from a fundamental vulnerability when applied to pretrained backbones. We also empirically demonstrate that pretraining priors enable models to circumvent UE-induced shortcuts and acquire true semantics.
- We propose BAIT, a bi-level optimization framework that binds perturbations to incorrect target labels, thereby disrupting the semantic-label mapping underpinned by pretraining priors and reconstructing the synthetic perturbation-label correlations.
- Extensive qualitative and quantitative experiments demonstrate the effectiveness and generalizability of BAIT against various pretrained backbones.

## 2 RELATED WORK

Recent studies have increasingly focused on societal implications, encompassing critical dimensions such as privacy (Li et al., 2025a; Zheng et al., 2025; Pu et al., 2025b), safety (Huang et al., 2020), and fairness (Xu et al., 2024). Unlearnable Examples (UEs) have emerged as a promising protection approach to prevent unauthorized data usage (Sandoval-Segura et al., 2023; Wang et al., 2025b). The goal is to inject imperceptible perturbations into data such that models trained on these perturbed samples achieve high training accuracy but fail to generalize to the clean test set, with test accuracy dropping to the chance level. To achieve this, prior studies mainly introduce "shortcuts" during training, guiding models to memorize synthetic perturbation-label associations instead of learning real semantics (Sandoval-Segura et al., 2022; Wu et al., 2023; Wang et al., 2025a). Recently, UEs have been extended to diverse realistic settings. Some works (He et al., 2023; Wang et al., 2024) explore unsupervised regimes, whereas UC (Zhang et al., 2023) and 14A (Chen et al., 2024) investigate label-agnostic scenarios and cross-dataset transferability, respectively. Additionally, VTG (Li et al., 2025b) offers a comprehensive assessment covering distribution shifts (Guo et al., 2024; Fang et al., 2025b; Xu et al., 2025a), novel classes, and input space discrepancies. Moreover, the applicability of UEs has extended beyond classification (Guo et al., 2023; Fang et al., 2025a) to other tasks, such as segmentation (Sun et al., 2024), point clouds (Zhu et al., 2024), and diffusion models (Zhao et al., 2024). These studies have greatly promoted the development of the UE literature.

However, existing efforts predominantly focus on applying UEs to randomly initialized backbones, which neglects the practical yet crucial case of pretrained models that underpin most modern applications, leaving a critical gap in the current UE literature. While some UE studies (Qin et al., 2023; Sun et al., 2024) utilize pretrained models for perturbation generation or auxiliary evaluation, they lack a systematic investigation into the vulnerability of UEs against pretrained models and offer no corresponding countermeasures. To the best of our knowledge, we make the first attempt to uncover the vulnerability of UEs against pretrained backbones and design a novel bi-level optimization framework to neutralize the influence of pretraining priors.

## 3 PROPOSED METHOD

In this section, we describe the UE setting targeting pretrained backbones and propose BAIT, a bi-level framework designed to bind perturbations to incorrect labels. Then, we elaborate on the implementation details and optimization strategies to construct effective UEs.

### 3.1 PROBLEM SETUP

In contrast to existing UE studies that target train-from-scratch classifiers, we introduce a new UE setting that focuses on rendering data unlearnable for pretrained classifiers. Such models possess rich prior knowledge from their pretraining data and thus are harder to be misled by the injected unlearnable perturbations. Below, we formulate the problem in the context of image classification.

**Objectives.** Let $(\boldsymbol{x}, y)$ be a labeled example, where $\boldsymbol{x} \in \mathcal{X}$ is a data point, and $y \in \mathcal{Y} = \{1, \ldots, K\}$ is its label. We denote the clean training and test sets as $\mathcal{D}_c$, $\mathcal{D}_t$, respectively. The goal of UEs is to transform the training set $\mathcal{D}_c$ into an unlearnable dataset $\mathcal{D}_u$ by injecting perturbations $\boldsymbol{\delta}$ to each example as $\boldsymbol{x}' = \boldsymbol{x} + \boldsymbol{\delta}$. Generally, the perturbation $\boldsymbol{\delta}$ is constrained by $\|\boldsymbol{\delta}\|_p \leq \epsilon$, ensuring imperceptibility to human vision, where $\| \cdot \|_p$ denotes the $L_p$ norm and $\epsilon$ is chosen to be sufficiently small. A pretrained classifier $f_\theta : \mathcal{X} \to \Delta_{\mathcal{Y}}$, where $\theta$ is the model parameter and $\Delta_{\mathcal{Y}}$ denotes the probability simplex over the label space $\mathcal{Y}$, is *fully finetuned* on $\mathcal{D}_u$. The effectiveness of perturbations $\boldsymbol{\delta}$ is then evaluated by measuring the performance degradation of $f_\theta$ on the clean test set $\mathcal{D}_t$, which would approximate the chance level if $\boldsymbol{\delta}$ can successfully counter the pretraining priors of $f_\theta$ and establish artificial perturbation-label correlations for the training process.

### 3.2 BINDING ARTIFICIAL PERTURBATIONS TO INCORRECT TARGETS

Existing works primarily focus on designing UEs to protect data from train-from-scratch classifiers. However, as shown in Figure 1, their unlearnability diminishes substantially when applied to pretrained backbones, where priors bypass spurious correlations and allow the model to learn real

data-label relationships. This fundamental vulnerability severely limits the practical deployment of UEs in protecting personal data from unauthorized exploitation.

To address the aforementioned limitation, we propose BAIT, a novel bi-level optimization framework[1] designed to bind artificial perturbations to incorrect targets that are semantically distinct from the ground truth, thereby deliberately diverting the learning process from genuine semantics. To achieve this, we devise a mislabel-perturbation binding mechanism that disrupts the inherent data-label alignment established by priors. This mechanism enforces class-wise perturbations to dominate the model's predictions, which steer the input to a designated incorrect target class regardless of the original image content. The formulation of BAIT is detailed as follows:

$$
\begin{aligned}
\min_{\boldsymbol{\delta}} \quad & \mathbb{E}_{(\boldsymbol{x},y)\sim\mathcal{D}_c}\mathcal{L}\big(f_\theta(\boldsymbol{x}^i + \boldsymbol{\delta}^j), y^j\big) \\
\text{s.t.} \quad & \theta \in \arg\min_\theta \mathbb{E}_{(\boldsymbol{x},y)\sim\mathcal{D}_c}\mathcal{L}\big(f_\theta(\boldsymbol{x}^i + \boldsymbol{\delta}^i), y^i\big),
\end{aligned}
\tag{1}
$$

where $\boldsymbol{x}$, $y$, and $\boldsymbol{\delta}$ are samples, labels, and class-wise perturbations, respectively; superscripts $i, j \in \{1, \ldots, K\}$ $(i \neq j)$ indicate distinct classes, with $K$ representing the total number of classes; function $f_\theta$ refers to a pretrained surrogate parameterized by $\theta$; $\mathcal{L}$ denotes the cross-entropy loss.

The inner and outer objectives of BAIT are intricately coupled to form a dynamic interplay. Specifically, (1) The inner optimization on model parameters aims to align the perturbed inputs with ground truth labels, fostering a standard semantic alignment. Concretely, a sample $\boldsymbol{x}$ from class $i$ is perturbed by the corresponding class-wise perturbation $\boldsymbol{\delta}^i$ and mapped to its true label $y^i$. (2) The outer optimization over perturbations actively breaks the data–label alignment established in the inner level and dynamically enforces intentional mislabel-perturbation bindings. Unlike prior UE methods (Huang et al., 2021; Ren et al., 2023; Fu et al., 2022) that preserve the original data-label correspondence ($\boldsymbol{x}^i + \boldsymbol{\delta}^i \to y^i$), we impose a more challenging cross-label assignment, explicitly mapping perturbed samples to designated incorrect target labels ($\boldsymbol{x}^i + \boldsymbol{\delta}^j \to y^j$). This mislabel-perturbation binding mechanism deliberately overrides the prior-driven data-label alignment, forcing models to rely on the spurious shortcuts induced by perturbations as the dominant training signal. Consequently, BAIT neutralizes the influence of pretraining priors, yielding unlearnable examples that inhibit pretrained models from capturing intrinsic semantics.

### 3.3 Optimization Strategy for Crafting Effective Unlearnable Examples

**Perturbation Generation.** We follow prior UE methods (Liu et al., 2024a; Chen et al., 2024) and develop a generator $\mathcal{G}$ to produce perturbations. $\mathcal{G}$ is designed with a standard encoder-decoder structure, which takes raw images $\boldsymbol{x}$ as input and generates perturbations as $\boldsymbol{\delta} = \mathcal{G}(\boldsymbol{x})$. To ensure visual imperceptibility, we constrain the perturbation magnitude by $\|\boldsymbol{\delta}\|_\infty \leq \epsilon$ with $\epsilon = 8/255$, in accordance with previous works (Huang et al., 2021; Ren et al., 2023; Liu et al., 2024a).

**Learning to Learn Unlearnable Perturbations.** Directly minimizing the full bi-level objective in Equation 1 is intractable. To overcome this challenge, we adopt a meta-learning strategy (Finn et al., 2017; Huang et al., 2020; Liu et al., 2024b), approximating the outer objective by unrolling the inner optimization for $N$ steps. Specifically, at the $n$-th iteration, let $\theta_n$ denote the surrogate model weights and $\boldsymbol{\delta}$ denote the perturbation (initialized to zero at the start of training). We create a copy of the current surrogate weights, $\theta'_{n,0} \leftarrow \theta_n$, which is then updated through $N$ inner-level optimization steps as follows:

$$
\theta'_{n,m+1} = \theta'_{n,m} - \alpha \nabla_{\theta'_{n,m}} \mathbb{E}_{(\boldsymbol{x},y)\sim\mathcal{D}_c}\mathcal{L}\big(f_\theta(\boldsymbol{x}^i + \boldsymbol{\delta}^i), y^i\big),
\tag{2}
$$

where $m \in \{0, 1, \ldots, N-1\}$, and $\alpha$ indicates the learning rate that controls the step size. This $N$-step unrolling enables a "look ahead" perspective during training, allowing us to explicitly assess how perturbations introduced at the current step gradually influence the outer mislabel-perturbation binding objective after $N$ inner updates.

---

[1]We note that while several previous studies (e.g., EMN (Huang et al., 2021), TUE (Ren et al., 2023)) refer to their formulations as bi-level optimization, they effectively implement a min-min optimization solved in an alternating manner. BAIT distinguishes itself by employing a formal bi-level structure, which allows for a more rigorous decoupling of the objectives to explicitly neutralize the influence of pretraining priors.

The unrolled surrogate model with weights $\theta'_{n,N}$ is then employed in the outer-level optimization to update perturbations, enforcing the perturbed samples onto incorrect target labels and misleading the intrinsic semantic-label connection within pretraining priors, which is shown below:

$$\boldsymbol{\delta}_{n+1}^j = \boldsymbol{\delta}_n^j - \beta \nabla_{\boldsymbol{\delta}_n^j} \mathbb{E}_{(\boldsymbol{x},y) \sim \mathcal{D}_c} \mathcal{L}(f_\theta(\boldsymbol{x}^i + \boldsymbol{\delta}_n^j), y^j), \tag{3}$$

where $i, j \in \{1, \ldots, K\}$ $(i \neq j)$ denote class indices; $\beta$ indicates the learning rate to optimize perturbations. In this procedure, each sample $\boldsymbol{x}$ from class $i$ is perturbed with $\boldsymbol{\delta}^j$ from a different class $j$ and forced towards the target incorrect label $y^j$, which optimizes perturbations that counter pretraining priors and prevent the model from relying on priors to associate samples with real labels.

The updated perturbations are incorporated into the corresponding samples $\boldsymbol{x}$ at the inner level, and the surrogate model $\theta_n$ is then updated accordingly to $\theta_{n+1}$, which serves as the initialization for the next iteration:

$$\theta_{n+1} = \theta_n - \alpha \nabla_{\theta_n} \mathbb{E}_{(\boldsymbol{x},y) \sim \mathcal{D}_c} \mathcal{L}(f_\theta(\boldsymbol{x}^i + \boldsymbol{\delta}^i), y^i). \tag{4}$$

The above meta-learning procedures are executed iteratively throughout the entire training process, ultimately yielding the optimized perturbations.

**Curriculum-Guided Target Label Selection.** To enhance the effectiveness of perturbations by directing perturbed samples toward diverse incorrect target labels, we dynamically select target labels rather than relying on fixed incorrect targets. Drawing inspiration from curriculum learning (Bengio et al., 2009; Wang et al., 2021), we design a three-stage easy-to-hard strategy according to the likelihood of misclassification, as shown below.

- **Stage 1: Hard Negative Classes**. We first select the non-ground-truth class with the highest logit score. Since these classes are most easily confused with the true class, they provide a natural starting point for optimization.
- **Stage 2: Random Classes**. We then randomly select non-ground-truth classes as the incorrect target class, increasing task difficulty and improving the generalizability of perturbations across varying target labels.
- **Stage 3: Most Dissimilar Classes**. Finally, we select the class with the lowest logit score. This constitutes the most challenging case, as perturbations must push predictions toward semantically unrelated classes.

This progressive curriculum learning strategy dynamically steers perturbations to perform mislabel-perturbation binding from easy to increasingly difficult targets, thereby enhancing their ability to mislead pretraining priors away from aligning samples with true labels. As a result, models are forced to depend on these perturbations for classification throughout training.

## 4 EXPERIMENTS

In this section, we first evaluate the effectiveness of the proposed optimization framework against pretrained backbones on standard benchmarks. We further examine the transferability across different pretraining priors, larger datasets, and network architectures. Moreover, we verify its efficacy in the conventional train-from-scratch setting and provide qualitative visualizations.

### 4.1 EXPERIMENTAL SETUP

**Datasets and Backbones.** Our experiments utilize standard benchmarks, including CIFAR-10, CIFAR-100 (Krizhevsky et al., 2009), and SVHN (Netzer et al., 2011). Beyond these, we extend our evaluation to Flowers102 (Nilsback & Zisserman, 2008) and a 100-class ImageNet subset (Deng et al., 2009) to assess performance in higher-resolution and more complex scenarios. Unless otherwise specified, evaluations are conducted on a fully trainable ImageNet-pretrained backbone with a learning rate of 0.001, in contrast to prior studies that utilize randomly initialized models. Backbones include ResNet-18 (He et al., 2016), ResNet-50 (He et al., 2016), VGG-11 (Simonyan & Zisserman, 2014), DenseNet-121 (Huang et al., 2017), GoogLeNet (Szegedy et al., 2015), Tiny-ViT (Wu et al., 2022), Swin Transformer Tiny (Liu et al., 2021), and ViT-B/16 (Dosovitskiy et al., 2021). The fully connected layer is modified to align with the class number of each downstream dataset. We employ

Table 1: Test accuracy (%) ↓ evaluation against ImageNet-pretrained backbones.

| Dataset | Backbone | Clean | EMN | TUE | REM | LSP | GUE | 14A | **Ours** |
|---------|----------|-------|-----|-----|-----|-----|-----|-----|----------|
| CIFAR-10 | ResNet-18 | 84.10 | 61.82 | 82.72 | 74.46 | 54.20 | 23.17 | 65.70 | **14.40** |
| | ResNet-50 | 86.48 | 57.54 | 85.68 | 85.06 | 65.00 | 71.42 | 72.81 | **14.82** |
| | VGG-11 | 88.39 | 80.90 | 87.67 | 63.32 | 55.63 | 45.77 | 50.80 | **22.14** |
| | DenseNet-121 | 87.17 | 63.31 | 86.30 | 61.05 | 62.66 | 72.31 | 74.89 | **20.32** |
| CIFAR-100 | ResNet-18 | 55.73 | 55.53 | 55.20 | 55.22 | 36.98 | 54.09 | 33.67 | **20.77** |
| | ResNet-50 | 60.73 | 45.78 | 59.54 | 58.69 | 42.84 | 53.14 | 36.58 | **28.02** |
| | VGG-11 | 63.62 | 38.10 | 62.52 | 35.25 | 39.26 | 57.02 | 30.58 | **26.64** |
| | DenseNet-121 | 61.92 | 49.94 | 61.14 | 48.37 | 36.27 | 56.78 | 44.34 | **25.81** |
| SVHN | ResNet-18 | 90.96 | 37.55 | 41.15 | 76.45 | 38.91 | 39.68 | 81.47 | **14.37** |
| | ResNet-50 | 92.09 | 30.29 | 35.52 | 87.38 | 45.48 | 91.74 | 85.59 | **18.86** |
| | VGG-11 | 93.34 | 56.24 | 77.54 | 51.50 | 49.89 | 32.14 | 82.89 | **17.75** |
| | DenseNet-121 | 92.97 | 58.22 | 40.53 | 71.87 | 53.56 | 73.34 | 86.83 | **11.61** |

classification accuracy on the clean test set as the evaluation metric, where lower accuracy signifies stronger unlearnability.

**Baselines.** We select six representative UE baselines, which are Error-Minimizing Noise (EMN) (Huang et al., 2021), Transferable Unlearnable Examples (TUE) (Ren et al., 2023), Robust Error-Minimizing Noise (REM) (Fu et al., 2022), Linear Separable Perturbation (LSP) (Yu et al., 2022), Game Unlearnable Example (GUE) (Liu et al., 2024a), and Universal Perturbation Generator (14A) (Chen et al., 2024).

**Implementation Details.** Our model is implemented with PyTorch and trained on a single NVIDIA L40S GPU. Unless specified otherwise, we utilize an ImageNet-pretrained ResNet-18 surrogate and set its learning rate $\alpha$ to 0.1. We adopt Adam (Kingma, 2014) with a learning rate $\beta$ of 0.001 to optimize the perturbation generator. Perturbations are constructed class-wise by averaging individual sample perturbations. The unrolling step is set to 1. For the three-stage curriculum learning strategy, we train each stage for 30 epochs on CIFAR-10 and CIFAR-100, and 20 epochs on SVHN. Perturbations are bounded by $8/255$ for imperceptibility. All results are reported as averages over five runs with different random seeds.

## 4.2 Unlearnability under Pretraining Priors

**Unlearnability against ImageNet-Pretrained Backbones.** To demonstrate the efficacy of BAIT in countering prior knowledge and generating effective perturbations, we conduct comprehensive evaluations against ImageNet-pretrained backbones. Perturbations are optimized and evaluated on the same backbone to ensure consistency. As shown in Table 1, our method outperforms current UE methods across all backbones and datasets. Specifically, it drives performance close to random guessing on CIFAR-10 and SVHN, and achieves sub-

Table 2: Test accuracy (%) ↓ comparison against re-implemented baselines with pretrained surrogates on an ImageNet-pretrained ResNet-18.

| Method | CIFAR-10 | CIFAR-100 | SVHN |
|--------|----------|-----------|------|
| EMN* | 59.84 | 45.20 | 32.33 |
| TUE* | 82.65 | 55.28 | 47.71 |
| REM* | 56.08 | 34.30 | 55.26 |
| **Ours** | **14.40** | **20.77** | **14.37** |

stantial accuracy reductions on CIFAR-100. Notably, even though 14A (Chen et al., 2024) incorporates a pretrained CLIP backbone to aid perturbation generation, our method still surpasses it on all benchmarks. These results highlight the superior capacity of BAIT to bind perturbations with designated incorrect labels, compelling models to rely on perturbations rather than priors and thereby preventing models from learning genuine semantics.

**Further Comparison to Re-implemented Baselines with ImageNet-Pretrained Priors.** Several UE methods do not exploit pretraining priors when generating perturbations. Therefore, a critical question is whether their deficiencies stem from this omission, and whether leveraging such priors would yield marked improvements. To rigorously examine this, we re-implement represen-

Table 3: Transferability across pretrained backbones with different priors. An ImageNet-pretrained ResNet-18 is used as the surrogate model for perturbation optimization, and the resulting unlearnable examples are evaluated against prior knowledge from CIFAR-10, CIFAR-100, and SVHN.

| Dataset | Pretraining Prior | EMN | TUE | REM | LSP | GUE | **Ours** |
|---|---|---|---|---|---|---|---|
| CIFAR-10 | CIFAR-100 | 48.58 | 82.26 | 60.88 | 60.64 | 27.05 | **20.58** |
| | SVHN | 28.33 | 49.00 | 37.91 | 38.53 | 15.55 | **9.75** |
| CIFAR-100 | CIFAR-10 | 33.67 | 61.05 | 24.77 | 35.08 | 66.93 | **22.89** |
| | SVHN | 62.57 | 62.18 | 16.77 | 26.14 | 59.10 | **11.21** |
| SVHN | CIFAR-10 | 19.13 | 12.80 | 70.43 | 26.73 | 29.35 | **11.27** |
| | CIFAR-100 | 28.75 | 19.47 | 64.42 | 47.71 | 24.14 | **17.49** |

Table 4: Test accuracy (%) ↓ evaluation against pretrained backbones on more complex datasets. Note that ImageNet$^*$ and ImageNet$^\dagger$ denote two randomly selected 100-class subsets of ImageNet with no overlapping classes.

| Dataset | Pretraining Prior | EMN | TUE | REM | LSP | GUE | **Ours** |
|---|---|---|---|---|---|---|---|
| Flowers | ImageNet | 44.36 | 46.63 | 43.91 | 47.76 | 45.18 | **24.91** |
| ImageNet$^*$ | ImageNet$^\dagger$ | 51.98 | 59.00 | 59.12 | 59.44 | 37.72 | **24.22** |

tative surrogate-based UE methods, replacing their randomly initialized surrogates with ImageNet-pretrained models to enable a fairer comparison. The revised variants are denoted as EMN$^*$, TUE$^*$, and REM$^*$, respectively. As shown in Table 2, even with access to ImageNet priors during perturbation optimization, our method consistently outperforms these baselines, exhibiting stronger capabilities to erode and disrupt pretrained knowledge. This advantage enables our unlearnable examples to reliably counter pretrained backbones in practical applications and to provide sustained protection against unauthorized data usage.

**Transferability across Diverse Pretraining Priors.** As introduced in Equation 1, we employ an ImageNet-pretrained surrogate during perturbation optimization and then evaluate the resulting unlearnable data on ImageNet-pretrained backbones. To consider practical scenarios involving diverse pretraining priors, we further examine the generalizability of BAIT across backbones pretrained on CIFAR-10, CIFAR-100, and SVHN. As shown in Table 3, BAIT clearly outperforms baselines and reduces the test accuracy substantially, demonstrating the efficacy of our UEs even when they are applied to backbones with different priors.

**Transferability to Larger Datasets.** We perform additional experiments to study the transferability of our perturbations to more complex datasets. We utilize CIFAR-10 for perturbation optimization and subsequently evaluate the performance on Flowers102 and an ImageNet subset. For Flowers102, we conduct evaluations against a standard ImageNet-pretrained ResNet-18. For ImageNet, to mitigate potential data leakage between the pretraining and finetuning phases, we partition ImageNet-1K into disjoint subsets. Specifically, we first randomly select 100 classes to construct the pretraining dataset, denoted as ImageNet$^\dagger$, which is utilized to pretrain a ResNet-18 backbone. Subsequently, we sample a distinct set of 100 classes (with no overlap with ImageNet$^\dagger$) to serve as the downstream data, denoted as ImageNet$^*$. From Table 4, we observe that our method clearly outperforms baselines, highlighting its superior transferability and effectiveness on more challenging scenarios.

**Cross-Architecture Unlearnability.** We further examine the transferability of unlearnability across network architectures. As shown in Figure 2, perturbations are optimized with a surrogate ResNet-18 and evaluated on both conventional CNN-based pretrained backbones (ResNet-50, VGG-11, DenseNet-121, and GoogLeNet) and advanced ViT-based pretrained backbones (Tiny-ViT, Swin Transformer Tiny, and ViT-B/16). The results on CIFAR-10 and CIFAR-100 consistently validate that BAIT effectively induces cross-architecture unlearnability, outperforming baselines by a distinct margin. We also observe that advanced ViT-based backbones possess stronger resilience to unlearnable data. For example, TUE retains over 90% test accuracy on CIFAR-10, and GUE exceeds 80% on CIFAR-100. These findings indicate that richer priors manifest superior resistance to the ap-

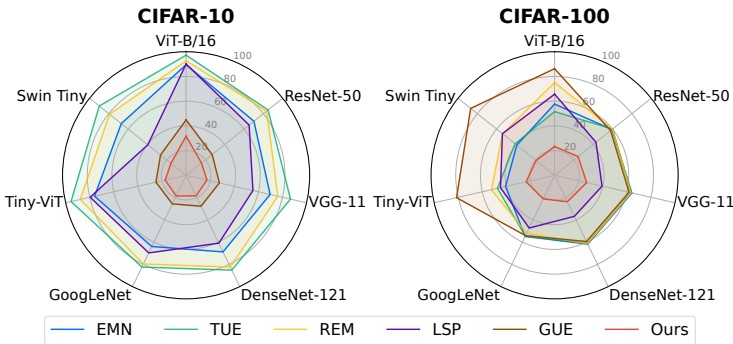

Figure 2: Evaluation of unlearnability transferability (test accuracy (%) ↓) across architectures, where UEs are generated using a ResNet-18 surrogate and evaluated against diverse pretrained backbones (including CNNs and ViTs).

Table 5: Ablation study of target label selection on ResNet-18. Stages 1, 2, and 3 denote selecting target labels from Hard Negative, Random, and Most Dissimilar Classes, respectively.

| Stage1 | Stage2 | Stage3 | CIFAR-10 | CIFAR-100 | SVHN |
|--------|--------|--------|----------|-----------|------|
| ✓ | | | 23.68 | 35.75 | 21.75 |
| ✓ | ✓ | | 20.60 | 23.94 | 16.58 |
| ✓ | ✓ | ✓ | **14.40** | **20.77** | **14.37** |

Table 6: Test accuracy (%) ↓ evaluation against randomly initialized backbones on CIFAR-10.

| Method | ResNet-18 | ResNet-50 | VGG-11 | DenseNet-121 |
|--------|-----------|-----------|--------|--------------|
| EMN | 16.42 | 13.45 | 16.93 | 14.71 |
| TUE | 10.67 | 12.23 | 12.79 | 17.60 |
| REM | 15.18 | 25.57 | 47.68 | 41.54 |
| LSP | 13.54 | 15.94 | 17.15 | 17.47 |
| GUE | 13.25 | 12.97 | 23.89 | 13.71 |
| 14A | 41.34 | 24.85 | 39.87 | 48.97 |
| **Ours** | **10.18** | **10.01** | **10.13** | **10.05** |

plied perturbations. Nevertheless, our method clearly surpasses baselines and successfully enforces unlearnability, thereby further demonstrating the efficacy of the proposed BAIT framework.

## 4.3 ABLATIONS AND FURTHER ANALYSES

**Impact of Curriculum-guided Target Label Selection.** The selection of target labels is crucial to establishing effective mislabel-perturbation bindings. To investigate the role of curriculum learning, we conduct rigorous ablation studies. As shown in Table 5, unlearnability against pretrained priors improves progressively with the inclusion of each stage. Specifically, we observe that selecting hard negative classes as the incorrect target labels (Stage 1) introduces the basic unlearnable effect, while incorporating random classes (Stage 2) and the most dissimilar classes (Stage 3) further strengthen it. These results highlight the contribution of each curriculum-guided stage and demonstrate that the synergistic integration of all three stages leads to the strongest performance.

**Unlearnability in the Conventional Train-from-Scratch Setting.** BAIT also provides a feasible solution when evaluating UEs with randomly initialized backbones. As shown in Table 6, we observe that while baseline methods demonstrate competitive unlearnability, our method exhibits superior efficacy and yields the lowest test accuracy. These results underscore the broad applicability of BAIT across both pretrained and randomly initialized backbones.

**Resistance to Defenses.** To evaluate the resilience against potential countermeasures, we apply diverse defense strategies, encompassing standard augmentations (Cutout (DeVries & Taylor, 2017),

Table 7: Test accuracy (%) ↓ evaluation against an ImageNet-pretrained ResNet-18 under additional defenses, where CIFAR-10 is utilized as the downstream dataset.

| Method | w/o | Cutout | CutMix | Mixup |
|--------|-----|--------|--------|-------|
| Clean | 84.10 | 83.25 | 83.06 | 84.14 |
| EMN | 61.82 | 51.60 | 54.71 | 68.70 |
| TUE | 82.72 | 81.54 | 81.35 | 82.69 |
| REM | 74.46 | 65.39 | 73.71 | 78.08 |
| LSP | 54.20 | 44.68 | 51.53 | 59.44 |
| GUE | 23.17 | 29.31 | 28.30 | 34.48 |
| **Ours** | **14.40** | **13.69** | **14.58** | **21.08** |

Table 8: Test accuracy (%) ↓ comparison against an ImageNet-pretrained ResNet-18 with JPEG compression defense on CIFAR-10.

| JPEG Compression | 90 | 80 | 70 | 60 | 50 | 40 | 30 | 20 | 10 |
|------------------|-----|-----|-----|-----|-----|-----|-----|-----|-----|
| Clean | 82.64 | 82.13 | 81.08 | 80.58 | 80.03 | 79.19 | 77.77 | 75.64 | 73.95 |
| EMN | 68.22 | 69.09 | 69.23 | 69.13 | 69.91 | 71.32 | 73.47 | 74.18 | 74.46 |
| TUE | 81.21 | 80.41 | 79.91 | 79.62 | 79.70 | 79.26 | 78.91 | 77.81 | 75.11 |
| REM | 77.15 | 77.29 | 78.08 | 78.17 | 78.34 | 78.55 | 78.33 | 78.14 | 75.33 |
| LSP | 55.47 | 57.24 | 59.56 | 61.43 | 63.48 | 63.84 | 64.86 | 68.53 | 71.00 |
| GUE | 60.94 | 65.59 | 66.73 | 66.86 | 68.39 | 69.18 | 70.11 | 72.33 | 72.88 |
| **Ours** | **15.91** | **16.47** | **16.70** | **16.96** | **17.12** | **18.93** | **20.41** | **27.48** | **68.08** |

CutMix (Yun et al., 2019), Mixup (Zhang et al., 2018)) and an advanced defense technique (JPEG compression (Liu et al., 2023)). As presented in Table 7, our method maintains reliable unlearnability across all transformations. Moreover, the results in Table 8 substantiate that BAIT clearly outperforms baselines, suppressing test accuracy to near-chance levels across a broad spectrum of JPEG compression qualities. While the unlearnability is impacted under extreme compression (Quality=10), given the pronounced visual artifacts introduced at this level, we argue that BAIT maintains exceptional resistance to JPEG defenses. Overall, these experiments demonstrate the superiority of BAIT in countering pretraining priors even when compounded by additional defenses.

**Visualization of Perturbed Samples.** The imperceptibility of the unlearnable perturbations to human vision serves as a key evaluation criterion for UEs, and we have explicitly constrained the magnitude of perturbations as $8/255$, in accordance with previous studies (Huang et al., 2021; Liu et al., 2024a). To provide a qualitative assessment, we visualize our perturbed images on CIFAR-10 and contrast them with existing UE methods. As illustrated in Figure 3, we observe that the perturbations optimized by BAIT are generally invisible to humans. We further present more visualizations of the perturbed images on CIFAR-100 and SVHN for a more comprehensive analysis, as shown in Appendix E.

**T-SNE Visualization.** We present t-SNE visualization (Maaten & Hinton, 2008) to further demonstrate the effectiveness of the unlearnable perturbations and their impact on models. As shown in Figure 4, while baseline methods are capable of inducing unlearnability in train-from-scratch backbones, manifested

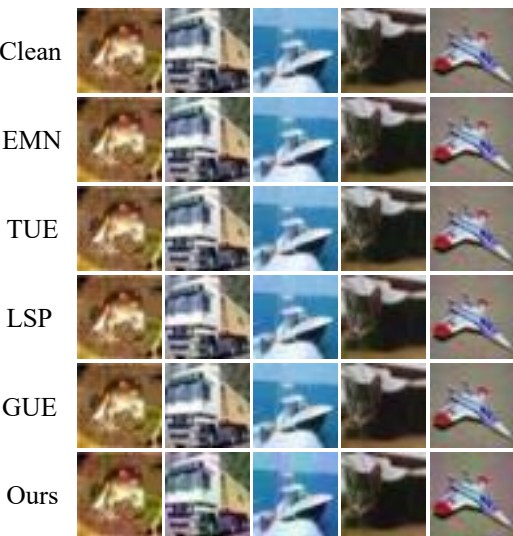

Figure 3: Perturbed examples on CIFAR-10.

as feature entanglement during testing, they fail to maintain this effect against pretrained models. In contrast, BAIT exhibits notable unlearnability across both randomly initialized and pretrained backbones, safeguarding authentic semantics and underscoring its superior effectiveness.

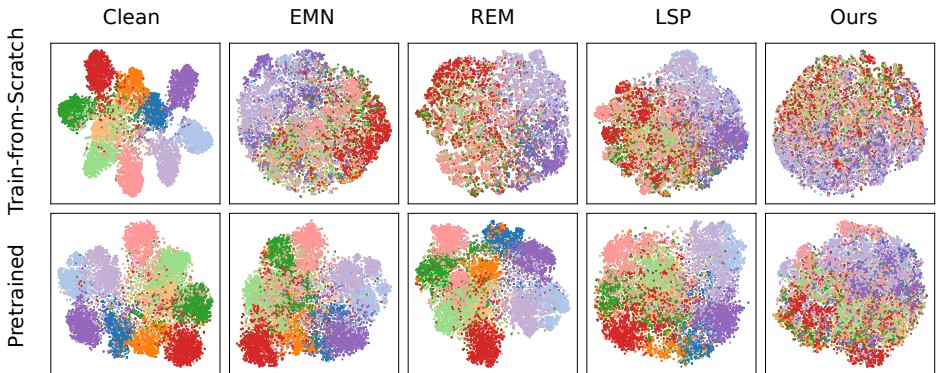

Figure 4: t-SNE visualization of the last layer features. Classifiers are trained on the perturbed training set and tested on the clean test set. The top row displays models trained in a train-from-scratch manner, whereas the bottom row shows models utilizing pretraining priors.

### 4.4 QUALITATIVE ANALYSIS ON THE OPTIMIZATION PROCESS OF BAIT

To provide further qualitative analyses on the optimization of perturbations, we use two variants of CIFAR-10 training samples to test the surrogate model, namely the perturbed training samples and the original clean training samples. During the optimization of BAIT, we plot the accuracy curve of the surrogate model $f_\theta$ on both types of samples, as illustrated in Figure 5. At the early stage, the surrogate model benefits from its inherent pretraining priors and acquires meaningful semantics, as reflected by the simultaneous improvement of performance on both perturbed and clean training images. However, as the optimization of perturbations, a divergence emerges: the accuracy on perturbed images keeps rising, while the accuracy on clean images progressively declines toward the chance level. This indicates that perturbations gradually mislead the real data-label association guided by pretraining priors and force the surrogate to rely on the spurious correlations between perturbations and labels, which prevents models from learning genuine semantics.

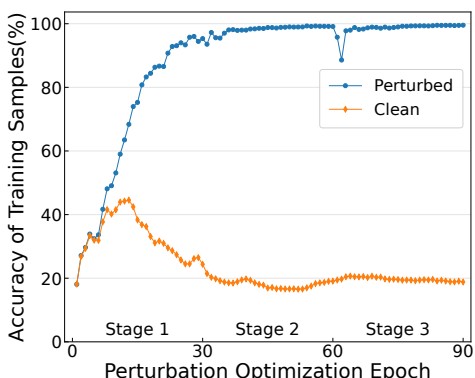

Figure 5: Training accuracy curve on CIFAR-10, illustrating that BAIT successfully misleads the ImageNet-pretrained surrogate model during perturbation optimization.

## 5 CONCLUSION

In this paper, we uncover an overlooked yet fundamental vulnerability of unlearnable examples that emerges when models are initialized with a pretrained backbone. We empirically show that the semantic-label pathway within priors enables models to bypass the shortcuts injected by UEs and acquire genuine semantics. To address this, we propose BAIT, a bi-level optimization framework designed to mislead pretraining priors. Specifically, the inner level imitates the standard prior-driven data-label alignment, while the outer level actively overrides it and establishes mislabel-perturbation bindings to enforce perturbed samples to align with designated incorrect targets. This mechanism counteracts pretraining priors and compels the model to rely on the spurious perturbation-label correlations. We integrate meta-learning and curriculum learning to facilitate optimization. Extensive experiments demonstrate that BAIT consistently outperforms state-of-the-art methods.

**Limitations.** While this paper reveals the vulnerability of UEs to pretraining priors in classification and proposes BAIT as a countermeasure, the extensibility of this approach to other downstream tasks, such as segmentation, remains relatively unexplored. Given the critical role of cross-task transferability, we consider this a pivotal direction for future investigation.

ETHICAL STATEMENT

This work is conducted in accordance with the ICLR Code of Ethics. Our study focuses on the design of unlearnable examples as a data protection mechanism, aiming to safeguard individuals' data from unauthorized exploitation in machine learning systems. All experiments are carried out on publicly available datasets under their respective licenses, with no involvement of human subjects or sensitive personal data. While techniques such as BAIT could, in principle, be misused for adversarial purposes, we emphasize that our intent is to strengthen privacy-preserving learning and to advance defenses against unauthorized model training. We report our methods and findings transparently to foster reproducibility and responsible use.

REPRODUCIBILITY STATEMENT

We have taken several measures to ensure the reproducibility of our work. All datasets used in our experiments are publicly available and described in detail in the main text. The implementation of the proposed BAIT framework, including model architectures, training schedules, hyperparameter settings, and evaluation procedures, is fully documented, and the source code has been released to facilitate the research community.

ACKNOWLEDGEMENTS

This work is supported by the Natural Sciences and Engineering Research Council of Canada (NSERC), Discovery Grants program.

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

APPENDIX: TABLE OF CONTENTS

## A   THE USE OF LARGE LANGUAGE MODELS (LLMS).

We declare that Large Language Models were used exclusively for language polishing, including correcting grammar, improving readability, and ensuring consistency in terminology. All core ideas, empirical demonstration, experimental design, and result analyses were entirely conceived and conducted by the authors.

## B   DISCUSSION ON PARAMETER UPDATES AND SEMANTIC LEARNING

### B.1   DETAILS OF THE PARAMETER UPDATES ILLUSTRATION

In Figure 1c, we present the total parameter updates when models are training on clean data and unlearnable data crafted by EMN and our method. We use the global maximum normalization to process the parameter update data, with details shown below.

Given the six groups of parameter changes measured during training, we use the L2 norm to calculate the parameter updates between epochs and denote the cumulative change of group $k$ at epoch $t$ as

$$v_t^{(k)} = \sum_{i=1}^{t} \|\Delta\theta_i^{(k)}\|_2, \tag{5}$$

where $\Delta\theta_i^{(k)}$ represents the parameter update at epoch $i$ for group $k$. Since $v_t^{(k)}$ is monotonically non-decreasing with respect to $t$, its maximum is attained at the final epoch $T$. Therefore, the global normalization factor is defined as

$$M = \max_{1 \le k \le 6} v_T^{(k)}. \tag{6}$$

The normalized cumulative change is then computed as

$$\tilde{v}_t^{(k)} = \frac{v_t^{(k)}}{M}, \quad \forall t = 1, \ldots, T, \ k = 1, \ldots, 6. \tag{7}$$

This normalization ensures that all curves are scaled into the interval $[0, 1]$, facilitating a fair comparison of their relative growth dynamics during training.

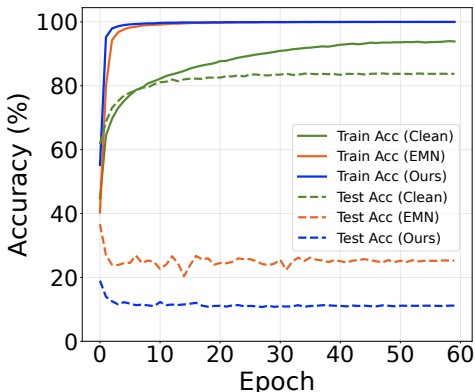

Figure 6: Learning curves illustration during standard UE evaluation on CIFAR-10 with ResNet-18, where training starts from a randomly initialized backbone.

### B.2   LEARNING CURVE OF TRAIN-FROM-SCRATCH BACKBONES

To further investigate the relationship between parameter update magnitude and model performance (Ye et al., 2025), and to validate that effective UEs suppress parameter updates and limit semantic acquisition, we visualize the accuracy curves of conventional UE evaluation that targets a randomly initialized model. As illustrated in Figure 6, we observe that both EMN and our method yield consistently low test accuracy. This observation aligns with the empirical findings in Figure 1c,

Table 9: The architecture of the perturbation generator.

| Block Name | Layer | Number |
|---|---|---|
| *Down-sampling layers* | | |
| Conv | Conv $(3 \times 3)$
InstanceNorm
ReLU | $\times 6$ |
| *Bottleneck layers* | | |
| Residual | ReflectionPad
Conv $(3 \times 3)$
BatchNorm
ReLU
ReflectionPad
Conv $(3 \times 3)$
BatchNorm | $\times 8$ |
| *Up-sampling layers* | | |
| ConvTranspose | ConvTranspose $(3 \times 3)$
InstanceNorm
ReLU | $\times 5$ |
| ConvTranspose | ConvTranspose $(6 \times 6)$
Tanh | $\times 1$ |

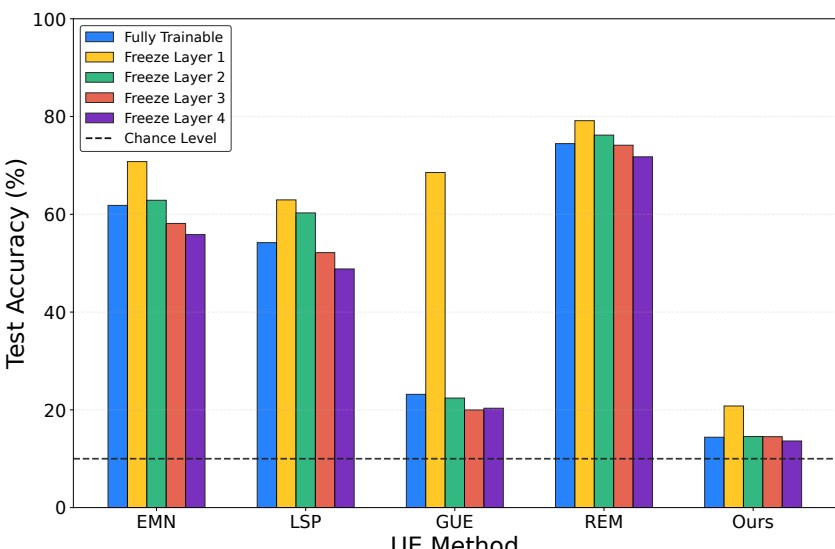

Figure 7: Effect of each pretraining layer of ResNet-18 against unlearnability on CIFAR-10.

where minimal parameter updates are recorded on train-from-scratch models. These results demonstrate that randomly initialized models fail to learn meaningful semantic knowledge and are misled by the UE-induced shortcuts.

## C  THE ARCHITECTURE OF PERTURBATION GENERATOR

In the main paper, we devise a generator to produce perturbations and craft unlearnable examples. We denote our perturbation generator as $\mathcal{G}$, which employs a standard encoder-decoder architecture. This structure comprises six down-sampling convolution layers, eight residual blocks (He et al., 2016), and six transposed convolution layers. The detailed architecture is shown in Table 9.

Table 10: Test accuracy change ($\Delta$Acc) on the original 1000-class ImageNet validation set when the ImageNet-pretrained model is finetuned on CIFAR-10.

| Finetuning Dataset | Accuracy Variation ($\Delta$Acc) | | | |
|---|---|---|---|---|
| | Epoch 1 | Epoch 2 | $\cdots$ | Epoch 60 |
| Clean CIFAR-10 | -36.45 | -53.83 | $\cdots$ | -55.28 |
| Unlearnable CIFAR-10 (Ours) | -40.66 | -55.11 | $\cdots$ | -55.35 |

## D    MORE EXPERIMENTS

### D.1    IMPACT OF PRETRAINING PRIORS AGAINST UNLEARNABILITY

To further examine the role of pretraining priors on unlearnability, we freeze each layer of an ImageNet-pretrained ResNet-18 model independently and analyze its specific impact on test accuracy. As shown in Figure 7, freezing layer 1 or layer 2 leads to an increase in test accuracy for several UE methods. This suggests that when low-level priors, which typically encode pixels, edges, and textures, are preserved from disruption by UEs, models are able to acquire more genuine semantics from the data. Notably, GUE exhibits a pronounced accuracy surge when the first layer is frozen, indicating that its perturbations mainly target shallow priors and thus become ineffective once these priors are fixed. In contrast, our method remains relatively effective even when shallow layers are frozen. Furthermore, freezing deeper layers, such as layer 3 and layer 4, results in a decrease in accuracy, suggesting that perturbations are now having no choice but to disrupt shallow priors and consequently achieve stronger unlearnability compared to the fully trainable pretrained scenario.

### D.2    PERFORMANCE VARIATION ON ORIGINAL PRETRAINING DATA DURING FINETUNING

To elucidate the interplay between downstream finetuning and pretraining priors, we conduct additional experiments to examine performance variation on the original source domain during finetuning. It is a well-known consensus in transfer learning that finetuning on distinct downstream tasks typically induces catastrophic forgetting, leading to performance decline on the original pretraining data (Lee et al., 2017; Kemker et al., 2018). To rigorously assess this, we report the relative performance variation when models are finetuned on either clean downstream data or our UEs. Specifically, we finetune an ImageNet-pretrained backbone on CIFAR-10 while monitoring the accuracy variation on the source ImageNet-1k dataset. As shown in Table 10, we observe that the magnitude of performance decline is comparable between models finetuned on clean data and those trained on our UEs. This indicates that the observed performance degradation is attributable to the intrinsic catastrophic forgetting, rather than being caused by the unlearnable perturbations. Consequently, we demonstrate that our method effectively protects downstream data privacy without exacerbating the risk of catastrophic forgetting or incurring additional damage to the pretrained knowledge within models.

### D.3    PERFORMANCE WITH A RANDOMLY INITIALIZED SURROGATE

To further evaluate the generalizability of the proposed method, we optimize BAIT with a randomly initialized surrogate model. We then evaluate its performance against the ImageNet-pretrained ResNet-18 model on three distinct datasets, including CIFAR-10, CIFAR-100, and SVHN. As shown in Table 11, our method outperforms baseline methods and exhibits strong capabilities in rendering data unlearnable, which further demonstrates its effectiveness against pretraining priors.

### D.4    CROSS-TASK TRANSFERABILITY

The transferability of unlearnable examples to other tasks (Fang et al., 2022; Zeng et al., 2024; Ji et al., 2024; Zhao et al., 2024; Zhou et al., 2025; Pu et al., 2025a) is crucial for practical applications. To this end, we conduct additional evaluations on the segmentation task, employing perturbations that are optimized on the CIFAR-10 classification dataset. We note that directly transferring perturbations may not be suitable since segmentation requires dense, pixel-wise predictions, whereas

Table 11: Test accuracy (%) ↓ evaluation against ImageNet-pretrained backbones, where BAIT utilizes a randomly initialized surrogate instead of a pretrained surrogate for perturbation optimization.

| Dataset | EMN | TUE | REM | LSP | GUE | 14A | **Ours** |
|---|---|---|---|---|---|---|---|
| CIFAR-10 | 61.82 | 82.72 | 74.46 | 54.20 | 23.17 | 65.70 | **15.97** |
| CIFAR-100 | 55.53 | 55.20 | 55.22 | 36.98 | 54.09 | 33.67 | **22.83** |
| SVHN | 37.55 | 41.15 | 76.45 | 38.91 | 39.68 | 81.47 | **17.93** |

Table 12: Evaluation of cross-task transferability on Pascal VOC 2012.

| Method | Pascal VOC 2012 Semantic (mIoU (%) ↓) | | | | |
|---|---|---|---|---|---|
| | Aeroplane | Bicycle | Bird | Boat | Bottle |
| Clean | 80.9 | 35.6 | 84.4 | 65.8 | 74.7 |
| UnSeg | **30.5** | **16.4** | **58.6** | **19.1** | **40.4** |
| LSP | 41.3 | 50.2 | 63.3 | 46.9 | 57.1 |
| **Ours** | 40.0 | 49.0 | 60.4 | 44.6 | 53.5 |

Table 13: Test accuracy (%) ↓ with different poisoning ratios against a ImageNet-pretrained ResNet-18 on CIFAR-10.

| Poison Ratio $r$ | 10% | 20% | 30% | 40% | 50% | 60% | 70% | 80% | 90% | 100% |
|---|---|---|---|---|---|---|---|---|---|---|
| Clean $(1-r)$ | 83.13 | 82.97 | 82.20 | 81.40 | 80.02 | 78.40 | 76.96 | 73.88 | 67.57 | — |
| EMN | 83.58 | 83.41 | 83.32 | 83.08 | 83.04 | 82.37 | 81.24 | 80.52 | 77.28 | 61.82 |
| TUE | 83.87 | 83.83 | 83.81 | 83.79 | 83.54 | 83.43 | 83.39 | 83.03 | 82.97 | 82.72 |
| REM | 84.18 | 83.79 | 83.63 | 83.05 | 82.69 | 82.90 | 81.96 | 81.09 | 79.81 | 74.46 |
| LSP | 83.56 | 83.51 | 82.53 | 82.49 | 81.48 | 80.29 | 79.68 | 77.97 | 74.91 | 54.20 |
| GUE | 83.94 | 83.58 | 83.48 | 82.48 | 81.79 | 81.14 | 79.88 | 78.18 | 74.50 | 23.17 |
| **Ours** | **83.06** | **82.94** | **81.67** | **81.51** | **81.11** | **79.43** | **78.13** | **74.95** | **71.88** | **14.40** |

classification relies on single, image-level labels. This discrepancy is likely to impact the effectiveness of UEs. Following the evaluation setup of UnSeg (Sun et al., 2024), we conduct experiments on the Pascal VOC 2012 dataset (Everingham et al., 2010). Note that UnSeg is specifically designed for segmentation and is considered to be a very strong baseline. We also evaluate the cross-task transferability of LSP (Yu et al., 2022) for comparison. As shown in Table 12, we observe that although not explicitly designed for cross-task transferability, our method can reduce the mIoU for several classes compared to clean training, indicating that our method maintains certain unlearnability when transferred to other downstream tasks. Moreover, our method outperforms LSP across five classes, demonstrating superior cross-task transferability compared with other classification-oriented UE baselines. We acknowledge that there is still room for improvement in bridging the performance between our method and the segmentation-specialist method UnSeg. Given the broad potential of cross-task UEs, we consider this as an important future research direction.

### D.5 VERIFICATION WITH DIFFERENT DATA PROTECTION RATIOS.

Although the standard UE setting applies perturbations to the entire training set, in practical scenarios, there may be only a portion of the data that is protected. To this end, following previous UE studies (Huang et al., 2021; Yu et al., 2022; Sadasivan et al., 2023), we also explore the impact of the data protection ratio on unlearnability. The experimental results are presented in Table 13. We observe that mixing clean samples with poisoned samples leads to a small performance increase compared to the clean training, indicating that models gain little semantic information from the UEs. Moreover, we observe that our method successfully reduces the test accuracy slightly compared to clean training at poison ratios of 10%, 20%, and 30%, and exhibits the smallest accuracy increase compared to baseline methods at other poison ratios, demonstrating its superior effectiveness.

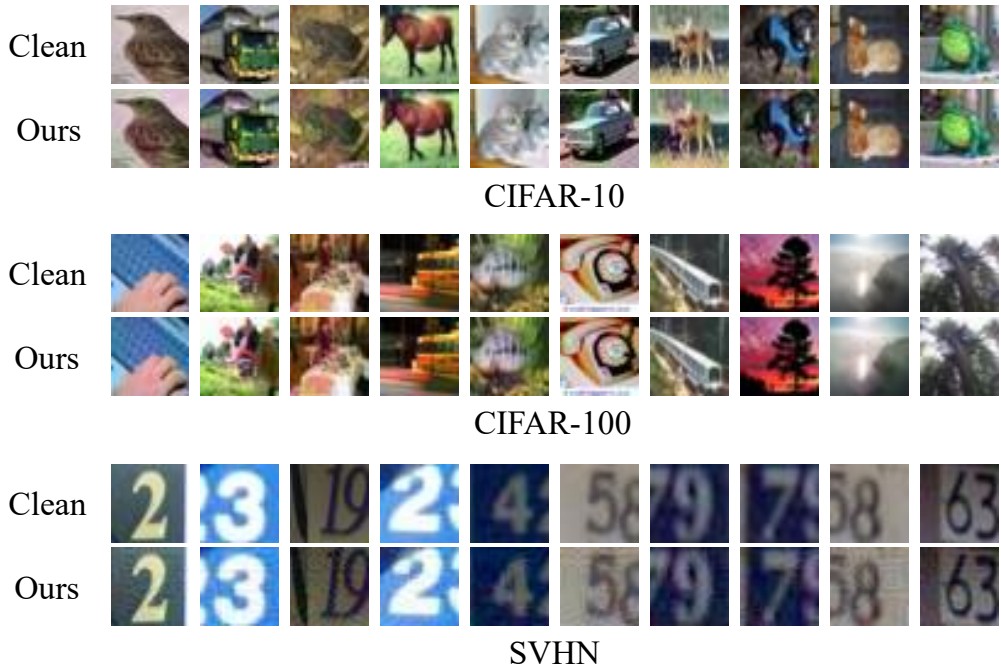

Figure 8: More visualizations of the unlearnable examples crafted by BAIT.

## E  MORE VISUALIZATIONS OF UNLEARNABLE EXAMPLES

To further ensure the imperceptibility of our unlearnable examples, we provide additional visualizations on CIFAR-10, CIFAR-100, and SVHN, as illustrated in Figure 8. Across all datasets, the perturbations remain visually subtle, generally exhibiting small magnitudes that do not degrade the perceptual quality of the images. These results further confirm that the crafted perturbations maintain imperceptibility to human observers while still enforcing unlearnability on models.

