# OpenReview forum: "When Priors Backfire: On the Vulnerability of Unlearnable Examples to Pretraining"
_ICLR.cc/2026/Conference — ICLR 2026 Poster_

### Official Review · Reviewer_rRcL · 2025-10-23

**Soundness:** 3
**Presentation:** 3
**Contribution:** 2
**Rating:** 4
**Confidence:** 4

**Summary:**

The paper introduces BAIT to solve the problem that existing UE methods fail when fine-tuning on clean pre-trained image-classification models by binding perturbations to incorrect target labels. Even though the proposed method seems to work only on small models (compared to the CV models mostly talked about nowadays) for image classification (which happens to be one of the easiest models to train from scratch for now), the results on common datasets look promising.

**Strengths:**

1. Despite some discussions in previous research, there are few solutions before to address the failure of UEs on pre-trained models.
2. The method description, experiment design, and the results seem convincing.
3. Despite some formatting issues, the writing and presentation are generally good.

**Weaknesses:**

1. The findings highlighted at the beginning of the paper that current UEs fail on pre-trained models do not seem to be that surprising, as this issue has been expressed in many previous studies.
2. From a practical perspective, the key to solving this problem, in my opinion, is to fine-tune a larger model using the additional data collected. Although this experiment will be more difficult, the authors should try to discuss the practical significance of this issue.
3. The paper doesn't seem to provide a thorough analysis or discussion of the relationship between pre-training data and fine-tuning data, or their impact on training, which is exactly what I'm most concerned about. If I perturb some data to prevent it from being trained and the model from learning, I'd first justify that the reduction in model performance (e.g., accuracy) is specific to the new data, and not to the original data. For example, a face recognition model might still accurately recognize the pre-trained data, but fail to recognize the perturbed data used in subsequent fine-tuning. This isn't a requirement for the author to verify this on a face recognition model, but I'd appreciate more in-depth analysis of this aspect.
4. The authors use three figures in Figure 1 to demonstrate the failure of existing UEs on pre-trained models. Figures 1a and 1b are convincing, but Figure 1c is not. Although PT and TS have significant differences in the dynamics of parameter updates, the authors appear to assume that parameter updates correspond to the model's actual learning of semantic knowledge from the data, a claim they haven't carefully justified. In fact, the relationship between the update amplitude of model parameters, training epochs and model performance is more complicated ([How far are we from true unlearnability?(ICLR'25)]).
5. The experimental part uses the Imagenet pre-trained model, but there is no experimental result on Imagenet and other larger datasets.
6. Although the authors discussed transferability, they only discussed the transferability of pre-trained models and architectures, and lacked discussion of the transferability of more important downstream tasks. Furthermore, lack of verification and discussion of the data protection ratio (i.e., the poisoning ratio).
7. Some papers also discussed the content related to UEs and pre-trained models, but the authors did not cite or mention them. [Learning the Unlearnable: Adversarial Augmentations Suppress Unlearnable Example Attacks(ICCV-AROW)] [UnSeg: One Universal Unlearnable Example Generator is Enough against All Image Segmentation(NeurIPS 2024)]
8. Some citation formatting issues

**Questions:**

Could the author focus on answering the first three concerns in the weakness section? It would be even better if the authors could provide some additional explanation or verification for the 4th point in weakness.

---

> ### Author Response · Authors · 2025-11-24
> **Response to Reviewer rRcL (1/5)**
>
> **Part 1/5**
>
> We thank reviewer rRcL for the constructive feedback and valuable suggestions. Below, we provide responses to each raised concern and question. Please feel free to reach out if any further clarification is required.
>
> ## **W1:** About findings on the vulnerability of UEs against pretrained models and relevant UE studies.
> We thank the reviewer for mentioning previous UE studies relevant to pretrained models. We respectfully clarify that although several UE papers have discussed UEs and pretrained models, to the best of our knowledge, **we make the first attempt to study the vulnerability of UEs against pretraining priors and design a novel optimization framework to address this limitation**. Regarding related papers mentioned by the reviewer, we provide a detailed explanation on how they incorporate pretrained models and highlight the difference between them and our paper as follows.
>
> - **Difference to [1]**. [1] mainly focuses on the defense of unlearnable examples. We notice that Section 4.8 of [1] demonstrates that pretrained models are able to gain accuracy on unlearnable data, a phenomenon consistent with our observation. However, **[1] only uses this phenomenon as a baseline to demonstrate the efficacy of its proposed defense**. In other words, there are **no investigation in [1] that actually explores this phenomenon**. In contrast, we take this observation as a starting point and conduct the first study to explore the vulnerability of UEs against pretraining priors.
> - **Difference to [2]**. To our understanding, **[2] mainly utilizes the pretrained model (SAM [3]) to assist the construction of UEs**. It does **not investigate** the impact of **pretraining priors** to UEs when training starts from a pretrained model, which is the specific core focus of our manuscript. Therefore, we argue that we make the first effort to reveal the vulnerability of UEs against pretraining priors.
>
> Despite the substantial differences between our paper and the aforementioned papers in incorporating pretrained models, we acknowledge that these papers are indeed relevant to the concept of both UEs and pretrained models, and we have incorporated corresponding **discussions** and **citations** in the **Related Work** section of our manuscript to improve the comprehensiveness. We would greatly appreciate it if the reviewer could kindly provide any reference to other UE studies that are relevant to pretrained models.
>
> ## **W2:** Practical significance of fine-tuning a larger model using the additional data collected.
> We value your comment and acknowledge that larger models intrinsically contain more robust knowledge and are likely to exhibit superior resistance to UEs. To demonstrate this, we have added additional evaluations against more **advanced** or **larger** models, including Tiny-ViT [4], Swin Transformer Tiny [5], and ViT-B/16 [6], on CIFAR-10 and CIFAR-100. As shown in **Table R3-1**, these models indeed exhibit **stronger robustness** when training with unlearnable data. For example, TUE achieves more than 90% test accuracy on CIFAR-10, and GUE obtains over 80% test accuracy on CIFAR-100. Even so, we observe that our method **outperforms baselines** clearly and still introduces unlearnability for these ViT models, which further demonstrates the efficacy of our proposed BAIT framework. We have added corresponding discussions in **Appendix F.2**.
>
> **Table R3-1.** Test accuracy (%) ↓ comparison against ImageNet-pretrained ViT models on CIFAR-10 and CIFAR-100 datasets.
> |  Dataset  |       Backbone        |  EMN  |  TUE  |  REM  |  LSP  |  GUE  |   Ours    |
> | :-------: | :-------: | :---: | :---: | :---: | :---: | :---: | :-------: |
> | CIFAR-10  |       Tiny-ViT        | 76.00 | 95.53 | 87.83 | 79.66 | 25.04 | **17.41** |
> |           | Swin Transformer Tiny | 67.09 | 89.90 | 79.23 | 39.62 | 26.45 | **15.67** |
> |           |       ViT-b/16        | 89.53 | 97.22 | 92.66 | 90.06 | 44.79 | **31.62** |
> | CIFAR-100 |       Tiny-ViT        | 40.72 | 47.47 | 52.39 | 44.93 | 81.40 | **23.64** |
> |           | Swin Transformer Tiny | 39.12 | 40.89 | 53.85 | 53.99 | 86.85 | **19.16** |
> |           |       ViT-b/16        | 57.63 | 51.53 | 75.09 | 65.68 | 86.26 | **23.30** |

---

> ### Author Response · Authors · 2025-11-24
> **Response to Reviewer rRcL (2/5)**
>
> **Part 2/5**
> ## **W3:** Discussion of the relationship between pre-training data and fine-tuning data.
> We appreciate your comment and would like to discuss the relationship between pretraining and fine-tuning data by monitoring the **performance variation** on **original** pretraining data during the **fine-tuning** on **new** data. Specifically, we conduct additional experiments to examine performance variations on the original data during fine-tuning. We note that, due to **catastrophic forgetting**, fine-tuning a pretrained model on downstream data is likely to cause a **sharp decline** in performance on the **original** data. To further investigate this, we report the relative performance change when models are fine-tuned on either clean downstream data or our UEs, with further discussion provided in **Appendix D.2**. Specifically, we fine-tune the pretrained model on CIFAR-10 while monitoring the variation in its performance on the original 1000-class ImageNet dataset compared to its performance before fine-tuning. As shown in **Table R3-2**, we observe that the accuracy reduction is **similar** for both **clean** and **UE** fine-tuning data. This indicates that the observed performance decrease arises from catastrophic forgetting rather than from the unlearnable perturbations. Therefore, we argue that our perturbations effectively protect downstream user data without compromising the model’s overall capability or increasing the risk of catastrophic forgetting during fine-tuning.
>
>  **Table R3-2.** Test accuracy change ($\Delta$ Acc) on original 1000-class ImageNet validation set when the ImageNet-pretrained model is finetuned on CIFAR-10.
> |     Fine-tuning Dataset     | Accuracy Variation in Epoch 1 | Accuracy Variation in Epoch 2 | ... | Accuracy Variation in Epoch 60 |
> | :-------------------------: | :---------------------------: | :---------------------------: | :-: | :----------------------------: |
> |       Clean CIFAR-10        |            -36.45             |            -53.83             | ... |             -55.28             |
> | Unlearnable CIFAR-10 (Ours) |            -40.66             |            -55.11             | ... |             -55.35             |
> ## **W4:** Further justification on parameter updates and actual semantic learning.
> Thanks for your comment. To further justify that effective UEs are able to reduce parameter updates and limit the acquisition of semantics, we illustrate the **training accuracy** and **testing accuracy** curves when pretrained models are fine-tuned on clean data and unlearnable data, as shown in **Appendix B.2** and **Figure 4**. Specifically, we are particularly interested in examining how the accuracy curve evolves when fine-tuning a **pretrained** model on UEs. As illustrated in **Appendix Figure 4a**, we observe that as the training accuracy of **EMN** improves, the testing accuracy maintains a **high** level of more than 60%, showing that the model fine-tuned on EMN-crafted unlearnable data actually **learns real semantics**. We also observe a similar phenomenon with a randomly initialized model in **Figure 4b**. This is consistent with our empirical findings in Figure 1(c), where the parameter updates of EMN are **larger** than those of our method and **close to clean training**. As for the accuracy curve of our method, the testing accuracy is actually decreasing to **chance level** as the training accuracy increases, demonstrating that models are **not** learning meaningful semantic knowledge. The parameter updates curve of our method in Figure 1(c) also stands for this claim, where it has fewer parameter updates compared to EMN and clean training.

---

> ### Author Response · Authors · 2025-11-24
> **Response to Reviewer rRcL (3/5)**
>
> **Part 3/5**
> ## **W5:** More Experiments on Larger Datasets.
> We add additional results on **Flowers** [7] and **ImageNet** to demonstrate the effectiveness of our method on larger datasets. For Flowers, we evaluate our perturbations against the standard ImageNet-pretrained ResNet-18. For ImageNet, to avoid overlap between the pretraining and fine-tuning datasets since both originate from ImageNet, we randomly select 100 classes from the full 1000-class ImageNet as the downstream fine-tuning data, denoted as ImageNet$^{\*}$. We then randomly select another 100 classes, with no overlap with ImageNet$^{\*}$, to pretrain a ResNet-18 model and use it during evaluation. This pretraining dataset is denoted as ImageNet$^{\dagger}$. As shown in **Table R3-3**, our method outperforms baselines with a clear margin, highlighting its effectiveness on the larger ImageNet dataset. We have included corresponding results and discussions in **Section 4.3 (Line 456-464)**.
>
> **Table R3-3.** Test accuracy (%) ↓ comparison against ImageNet-pretrained backbones on Flowers and ImageNet subset. Note that ImageNet$^{\*}$ and ImageNet$^{\dagger}$ denote two randomly selected 100-class subsets of ImageNet with no overlapping classes.
> | Downstream Dataset |  Pretraining Prior   |  EMN  |  TUE  |  REM  |  LSP  |  GUE  |   Ours    |
> | :----------------: | :------------------: | :---: | :---: | :---: | :---: | :---: | :-------: |
> |      Flowers       |       ImageNet       | 44.36 | 46.63 | 43.91 | 47.76 | 45.18 | **24.91** |
> |   ImageNet$^{\*}$   | ImageNet$^{\dagger}$ | 51.98 | 59.00 | 59.12 | 59.44 | 37.72 | **24.22** |

---

> ### Author Response · Authors · 2025-11-24
> **Response to Reviewer rRcL (4/5)**
>
> **Part 4/5**
> ## **W6:** Transferability to other tasks and verification of the data protection ratio.
> We provide additional evaluations on downstream tasks and data protection ratio as follows.
> - **Transferability to other downstream tasks.**
> 	- We thank the reviewer for the suggestion of evaluating the transferability to downstream tasks. However, to the best of our knowledge, we are not aware of existing UE works that study the transferability of UEs across tasks, such as from classification to segmentation or detection. We would greatly appreciate it if the reviewer could kindly provide any guidance on this.
> 	- Even so, we conduct evaluations on the transferability of our method to a different image **segmentation** task, with perturbations optimized from the CIFAR-10 classification dataset. We note that **directly transferring** perturbations may **not** be suitable since segmentation requires dense, pixel-wise predictions, whereas classification relies on single, image-level labels. This discrepancy is likely to impact the effectiveness of UEs. Following the evaluation setup of UnSeg [2], we conduct experiments on **Pascal VOC 2012** [8]. Note that UnSeg is specifically designed for segmentation and is considered to be a very strong baseline. We also evaluate the cross-task transferability of LSP for comparison. As shown in **Table R3-4**, we observe that although not explicitly designed for cross-task transferability, our method is able to **reduce** the mIoU for several classes compared to clean training, indicating that our method **maintains certain unlearnability** when transferred to other downstream tasks. Moreover, our method **outperforms** LSP across five classes, demonstrating superior cross-task transferability compared with other classification-oriented UE baselines. We have included the above corresponding discussion in **Appendix F.3**. We also note that there is still room for improvement in bridging the performance between our method and the segmentation-specialist method UnSeg. Given the broad potential of cross-task UEs, we consider this as an important **future** research direction and have added corresponding discussions in the **Conclusion** section.
>
> **Table R3-4.** Evaluation of cross-task transferability from classification to segmentation on Pascal VOC 2012.
> | Method / mIoU (%) ↓ | Aeroplane | Bicycle  |   Bird   |   Boat   |  Bottle  |
> | :-----------------: | :-------: | :------: | :------: | :------: | :------: |
> |        Clean        |   80.9    |   35.6   |   84.4   |   65.8   |   74.7   |
> |        UnSeg        | **30.5**  | **16.4** | **58.6** | **19.1** | **40.4** |
> |         LSP         |   41.3    |   50.2   |   63.3   |   46.9   |   57.1   |
> |        Ours         |   40.0    |   49.0   |   60.4   |   44.6   |   53.5   |
>
> - **Verification of data protection ratio.**
> 	- We verify the impact of poison ratios on unlearnable examples and present the results in **Table R3-5**. We observe that mixing clean samples with poisoned samples leads to a small performance increase compared to the clean training, indicating that models gain **little** semantic information from the UEs. Moreover, we observe that our method even **reduces** the test accuracy slightly compared to clean training at poison ratios of 10\%, 20\%, and 30\%, and exhibits the **smallest accuracy increase** compared to baseline methods at other poison ratios, demonstrating its superior effectiveness. The corresponding analyses have been added in **Appendix F.4**.
>
> **Table R3-5.** Effectiveness under different poison ratios against the ImageNet-pretrained ResNet-18 model on CIFAR-10.
> | Poison Ratio r |    10%    |    20%    |    30%    |    40%    |    50%    |    60%    |    70%    |    80%    |    90%    |   100%    |
> | :------------: | :-------: | :-------: | :-------: | :-------: | :-------: | :-------: | :-------: | :-------: | :-------: | :-------: |
> |  Clean (1-r)   |   83.13   |   82.97   |   82.20   |   81.40   |   80.02   |   78.40   |   76.96   |   73.88   |   67.57   |     \     |
> |      EMN       |   83.58   |   83.41   |   83.32   |   83.08   |   83.04   |   82.37   |   81.24   |   80.52   |   77.28   |   61.82   |
> |      TUE       |   83.87   |   83.83   |   83.81   |   83.79   |   83.54   |   83.43   |   83.39   |   83.03   |   82.97   |   82.72   |
> |      REM       |   84.18   |   83.79   |   83.63   |   83.05   |   82.69   |   82.90   |   81.96   |   81.09   |   79.81   |   74.46   |
> |      LSP       |   83.56   |   83.51   |   82.53   |   82.49   |   81.48   |   80.29   |   79.68   |   77.97   |   74.91   |   54.20   |
> |      GUE       |   83.94   |   83.58   |   83.48   |   82.48   |   81.79   |   81.14   |   79.88   |   78.18   |   74.50   |   23.17   |
> |    **Ours**    | **83.06** | **82.94** | **81.67** | **81.51** | **81.11** | **79.43** | **78.13** | **74.95** | **71.88** | **14.40** |

---

> ### Author Response · Authors · 2025-11-24
> **Response to Reviewer rRcL (5/5)**
>
> **Part 5/5**
> ## **W7:** Missing citation of relevant UE studies.
> We thank the reviewer for mentioning UE studies relevant to pretrained models, and we have added discussions and citations in the **Related Work** Section (**Line 119-122**).
>
> ## **W8:** Some citation formatting issues.
> We value your comment and have corrected all citation formatting issues in our manuscript.
>
> ## **Reference**
>
> [1] Learning the Unlearnable: Adversarial Augmentations Suppress Unlearnable Example Attacks. Qin et al., ICCV-AROW 2023.
>
> [2] UnSeg: One Universal Unlearnable Example Generator is Enough against All Image Segmentation. Sun et al., NeurIPS 2024.
>
> [3] Segment Anything. Kirillov et al., ICCV 2023.
>
> [4] TinyViT: Fast Pretraining Distillation for Small Vision Transformers. Wu et al., ECCV 2022.
>
> [5] Swin Transformer: Hierarchical Vision Transformer using Shifted Windows. Liu et al., ICCV 2021.
>
> [6] An Image is Worth 16x16 Words: Transformers for Image Recognition at Scale. Dosovitskiy et al. ICLR 2021.
>
> [7] Automated Flower Classification over a Large Number of Classes. Nilsback et al., ICVGIP, 2008.
>
> [8] The Pascal Visual Object Classes (VOC) Challenge. Everingham et al. IJCV 2010.

---

### Official Review · Reviewer_vgQM · 2025-10-29

**Soundness:** 2
**Presentation:** 2
**Contribution:** 3
**Rating:** 4
**Confidence:** 5

**Summary:**

This paper asks whether unlearnable examples are effective when training begins from a pretrained model. The authors find that error-minimizing noise, transferable unlearnable examples, robust error-minimizing noise, linearly separable perturbations, and a few other unlearnable examples cannot consistently reduce the test accuracy over the course of training. The authors propose optimizing class-wise, error-minimizing perturbations that make it so that when someone trains a new model starting from pretrained weights, the test accuracy can be reduced.

**Strengths:**

Based on Figure 1 (a) it appears that the proposed class-wise perturbations reduce the test accuracy of both pretrained and trained-from-scratch models, whereas other unlearnable example methods only reduce test accuracy for train-from-scratch models. I also agree that for unlearnable examples the case of utilizing pretrained weights is much more likely (and a more realistic scenario) than train-from-scratch, so unlearnable examples must be able to hold up to this approach (please see weaknesses 3. on how to properly see whether the current method holds up to defenses).

Starting from ImageNet pretrained models, Table 1 shows that their unlearnable examples reduce test accuracy by a wide margin. Their perturbations can prevent training (by reducing accuracy to nearly chance) on CIFAR-10, CIFAR-100, and SVHN.

Because unlearnable examples like [2] use a surrogate for a particular dataset (i.e. CIFAR-10), the authors consider a fair comparison where they re-implement [2] but using an imagenet-pretrained model as the surrogate. Here, they again find that their approach is better.

**Weaknesses:**

1. The "core innovation" (L180-181) is an error-minimizing noise, which is explored in [2]. Technically, the original error-minimizing noise of [2] also "binds perturbations to designated incorrect labels that are semantically different from the ground truth, deliberately steering learning away from genuine semantics." (L157-158) because [2] uses a pretrained surrogate model. By using a pretrained surrogate, the perturbations are like features of the pretrained model.

2. I think Eq. 1 does not line up with what the authors are proposing. The authors state that "the inner level mirrors the standard UE objective by injecting perturbations that discourage the model from encoding genuine semantic" implying the optimization is over the perturbation but the inner "s.t." objective says that we seek to find a theta (model params) that minimizes the expected loss (train a model on perturbed images). The $\theta_i$ probably requires an additional "level" of optimization because those aren't fixed.

3. Line 427 "Resistance to defense strategies". The only defense that should be considered is ISS [1] : I am particularly interested in an evaluation of *just ISS* (different JPEG compression qualities must be tried: 0.9, 0.8, 0.7, etc) instead of all the other "defenses" (cutout, cutmix, mixup, etc.) because JPEG has been shown to be so effective. This paper broke a number of existing "unlearnable datasets" but this submission only considers cutout, cutmix, mixup (which can no longer be considered a defense after the ISS paper). Additionally, being a class-wise perturbation, I recommend looking at the orthogonal projection [3] defense (Section 4.4 of [3]). They argue class-wise perturbations can be easily broken.

4. Line 249: "we evaluate the effectiveness of the proposed optimization framework against pretrained backbones on standard benchmarks". The paper doesn't specify what evaluation means exactly. Do you mean that we finetune only the last linear layer of a model? If so, does the last linear layer starting from random weights or the pretrained weights? Alternatively, do you mean that we finetune all model parameters? Based on Eq. 2 it seems the answer is finetune all model parameters starting from pretrained model parameters, but this has to be stated clearly in words somewhere. Using the phrase "craft effective UEs against pretrained backbones" (L083) makes it sound like only the last layer is finetuned.

[1] Image shortcut squeezing: Countering perturbative. Liu et al. 2023

[2] Unlearnable Examples: Making Personal Data Unexploitable. Huang et al., 2021

[3] What Can We Learn from Unlearnable Datasets?. Sandoval-Segura et al., 2023

Minor:
- Acronyms are used before they are defined: L072 has "EMN" but defined until L267. Please check if there are other instances of acronyms.

**Questions:**

1. If the authors believe their statement of Eq. 1 is correct, can they explain it in more detail to an undergraduate? How does it line up with the steps they propose in L176-188? Ideally follow the format of the last sentence of pg.3 of [2] where they say "this is a min-min bi-level optimization: the inner minimization...finds the bounded noise that...while the outer minimization finds the parameters that..."
2. For L313 "Reimplementeed baselines with imagenet-pretrained priors", did you use a surrogate model that was finetuned on the target dataset? For example, for generating CIFAR-10 perturbations, did you first finetune the ImageNet-pretrained model on CIFAR-10 or did you use the ImageNet-pretrained weights directly as the surrogate?

---

> ### Author Response · Authors · 2025-11-24
> **Response to Reviewer vgQM (1/3)**
>
> **Part 1/3**
>
> We thank reviewer vgQM for the constructive feedback and valuable suggestions. Below, we provide responses to each raised concern and question. Please feel free to reach out if any further clarification is required.
> ## **W1:** Major difference between error-minimizing noise of [1] and our method.
> We thank the reviewer for highlighting the conceptual connection to the error-minimizing noise in [1]. However, there are some misunderstandings and we would like to clarify that there are **fundamental distinctions** between error-minimizing noise [1] and our method. Below, we delineate major differences from three aspects: **different data-label association**, **different formulation** and **different optimization strategy**.
> - **Different Data-Label Association.** We clarify that the goal of our method is to find perturbation $\delta$ that links input samples with **incorrect labels**, while error-minimizing noise [1] focuses on mapping samples with **ground truth labels**. Specifically, the **error-minimizing noise** of [1] aims at finding noises $\delta$ that minimizes the classification loss in the inner minimizaition while finding parameters $\theta$ that also minimize the model's classification loss in the outer minimization. Both minimization are establishing the **same** **correct** data-label associations. **In contrast**, our rationale is to **counter** the data-label alignment within pretraining priors (as noted in **Line 160-162**). To achieve this, we aim to bind perturbations to **designated** **incorrect** labels and deliberately pushes learning away from real semantics. Therefore, our optimization goal and perturbation-label association is intrinsically different to the error-minimizing noise [1].
> - **Different Formulation.** We clarify that our formulation (Eq.1) is a **true** bi-level optimization problem and is fundamentally different from the standard min-min optimization of [1] (**not** a true bi-level optimization problem). As noted in [2, 3], **standard min-min optimization** cooperates variables to minimize a **single shared** objective. In contrast, **bi-level optimization** involves a hierarchical structure where the outer and inner levels optimize **distinct objectives** that **mutually influence** each other, making the outer solution strictly dependent on the inner level's optimal response to a different goal. Given this, although error-minimizing noise is described in [1] as a min-min bi-level optimization problem, its inner level (optimization of perturbation $\delta$) and outer level (optimization of $\theta$) share an **identical objective** to minimize the same classification loss. Therefore, to our understanding, it is mathematically a standard min-min optimization problem, which we will further verify by discussing its official optimization method in the next point. Unlike [1], the inner and outer objectives of our formulation are **different** and **mutually influenced**, where the inner level optimize model parameters $\theta$ to associate $x^i$ with its **ground truth** label $y^i$, while the outer level optimizes perturbations $\delta$ to bind $x^i$ with the **designated incorrect** label $y^j$. This makes our formulation a true bi-level optimization problem and requires a specific optimization strategy to solve it.
> - **Different Optimization Strategy.** Due to the major difference between standard min-min optimization and bi-level optimization, we employ a different optimization strategy compared to [1] to solve our formulation. Specifically, for error-minimizing noise [1] that shares the same inner and outer objective, it solves it by simply updating parameters $\theta$ and noise $\delta$ **alternatingly** (as noted in its official code). In contrast, our bi-level objective can not be solved by alternating update parameters, and we adopt **meta-learning** [2, 4] to approximate the outer objective by **unrolling the inner optimization** (as noted in our Supplementary code).
> - Moreover, we find that the official code of [1] shows that it incorporates a **randomly initialized** surrogate model during its min-min optimization instead of using a pretrained surrogate model. We also find that the **pretrained** surrogate model is actually used to build the comparison baseline "**Error-Maximizing Noise**" (as noted in page 4 of [1]). We would greatly appreciate it if the reviewer could kindly provide any additional guidance on the use of pretrained model in building the error-minimizing noise of [1].

---

> ### Author Response · Authors · 2025-11-24
> **Response to Reviewer vgQM (2/3)**
>
> **Part 2/3**
>
> ## **W2:** Clarification on perturbation and model parameter optimization.
> We apologize that our statement brings misunderstandings and would like to clarify that the **perturbations $\delta$** are **solely** **optimized** in the **outer level**. For the **inner level**, **optimization is conducted** **only for model parameters $\theta$**, where perturbations $\delta$ are added to input $x$ and aim to trick the model to rely on fixed perturbations instead of learning real semantics for prediction. We have revised the statement regarding the inner level in the **Introduction Section (Line 87-90)** and **Section 3.2 (Line 183-186)** to better present our optimization procedure.
>
> ## **W3:** Additional defense evaluations of JPEG and orthogonal projection.
> We value your suggestion and add additional defense evaluations of JPEG defense [5] and Orthogonal Projection defense [6]. Both evaluations are conducted on CIFAR-10 with an ImageNet-pretrained ResNet-18 model. As shown in **Table R2-1**, our method outperforms baseline methods across all **JPEG defenses** with different compression quality levels 90, 80, and 70, etc (corresponding to the reviewer’s notation 0.9, 0.8, and 0.7, etc.). We observe that our method reduces clean test accuracy greatly and approximates **random guessing level** until the JPEG quality is set to 10, while baseline methods fail to introduce unlearnability even when the JPEG quality is set to higher values like 90. Considering images would be distorted visibly with a strong JPEG compression quality of 10, we argue that our method exhibits exceptional resistance against the JPEG defense. Moreover, as shown in **Table R2-2**. we observe our method exhibits remarkable performance compared to baselines when applying **orthogonal projection defense** to learn from the unlearnable data. In summary, these defense evaluations further demonstrates the superior robustness of our proposed method, and we have included the results in **Table 8** and **Table 9** of **Section 4.3** in our manuscript.
>
> **Table R2-1.** Defense evaluations with JPEG compression against ImageNet-pretrained ResNet-18 model on CIFAR-10.
>
> | JPEG Compression |    90     |    80     |    70     |    60     |    50     |    40     |    30     |    20     |    10     |
> | :--------------: | :-------: | :-------: | :-------: | :-------: | :-------: | :-------: | :-------: | :-------: | :-------: |
> |      Clean       |   82.64   |   82.13   |   81.08   |   80.58   |   80.03   |   79.19   |   77.77   |   75.64   |   73.95   |
> |       EMN        |   68.22   |   69.09   |   69.23   |   69.13   |   69.91   |   71.32   |   73.47   |   74.18   |   74.46   |
> |       TUE        |   81.21   |   80.41   |   79.91   |   79.62   |   79.70   |   79.26   |   78.91   |   77.81   |   75.11   |
> |       REM        |   77.15   |   77.29   |   78.08   |   78.17   |   78.34   |   78.55   |   78.33   |   78.14   |   75.33   |
> |       LSP        |   55.47   |   57.24   |   59.56   |   61.43   |   63.48   |   63.84   |   64.86   |   68.53   |   71.00   |
> |       GUE        |   60.94   |   65.59   |   66.73   |   66.86   |   68.39   |   69.18   |   70.11   |   72.33   |   72.88   |
> |     **Ours**     | **15.91** | **16.47** | **16.70** | **16.96** | **17.12** | **18.93** | **20.41** | **27.48** | **68.08** |
>
> **Table R2-2.** Defense evaluation with Orthogonal Projection Defense against ImageNet-pretrained ResNet-18 model on CIFAR-10.
>
> |  Method  | Orthogonal Projection Defense |
> | :------: | :---------------------------: |
> |   EMN    |             43.89             |
> |   TUE    |             70.47             |
> |   REM    |             67.38             |
> |   LSP    |             55.06             |
> |   GUE    |             49.37             |
> | **Ours** |           **29.11**           |
>
> ## **W4:** Clarification on specific fine-tuning setup.
> Thanks for your comment. We are indeed fine-tuning **all model parameters** on unlearnable data and studying whether UEs remain effective when training starts from a pretrained model, as described in the Problem Setup section (**Line 146-147**). To specify our setting more clearly, we have also revised **Line 085**, **Line 259** and other descriptions in our manuscript accordingly.

---

> ### Author Response · Authors · 2025-11-24
> **Response to Reviewer vgQM (3/3)**
>
> **Part 3/3**
> ## **MW1:** Proper definition of acronyms.
> We appreciate your comment and have carefully revised the manuscript (**Line 66-68**) to ensure that every acronym is properly defined before its first use.
>
> ## **Q1:** More explanation of our bi-level optimization framework.
> We thank the reviewer for the suggestion. We provide an explanation following the format of [1] to clarify our bi-level formulation (Eq. 1) as follows:
>
> - Our formulation is a min-min bi-level optimization problem with **different goals** at each level, where the inner level optimizes model parameters $\theta$ to align the input $x^i$ with the **ground truth** label $y^i$ by adding corresponding perturbation $\delta^i$, then the outer level enforces **cross-label** assignment by adding perturbation $\delta^j$ to bind the **same** input $x^i$ onto a **designated** **incorrect** label $y^j$.
> - This mislabel-perturbation binding mechanism steers the pretrained model away from establishing real data-label associations and compels the model to rely on the spurious perturbation-label correlations. In this way, perturbations are able to actively counter pretraining priors.
>
> We have revised our paper in the **Section 3.2 (Line 179-182)** accordingly to present it more clearly.
>
> ## **Q2:** Specific utilization manner of ImageNet-pretrained surrogate models.
> We value your comment and clarify that we **directly** use the ImageNet-pretrained model as the surrogate to re-implement baseline methods. The reasons are two-fold.
> - Our method also directly leverages an ImageNet-pretrained surrogate model during perturbation optimization. Therefore, it is **in accordance with our method** and could provide a fair comparison.
> - Fine-tuning an ImageNet-pretrained model would add extra procedures and **increase computational complexity** for generating UEs, limiting their practical applications. Therefore, we adopt the publicly available ImageNet-pretrained model instead of fine-tuning it on the target dataset.
>
> ### **Reference**
> [1] Unlearnable Examples: Making Personal Data Unexploitable. Huang et al., ICLR 2021.
>
> [2] Bilevel Programming for Hyperparameter Optimization and Meta-Learning. Franceschi et al., ICML 2018.
>
> [3] A Review on Bilevel Optimization: From Classical to Evolutionary Approaches and Applications. Sinha et al., IEEE Transactions on Evolutionary Computation 2017.
>
> [4] Model-agnostic meta-learning for fast adaptation of deep networks. Finn et al., ICML  2017.
>
> [5] Image Shortcut Squeezing: Countering Perturbative Availability Poisons with Compression. Liu et al. 2023.
>
> [6] What Can We Learn from Unlearnable Datasets?. Sandoval-Segura et al., NeurIPS 2023.

---

### Official Review · Reviewer_TKJA · 2025-10-30

**Soundness:** 3
**Presentation:** 3
**Contribution:** 3
**Rating:** 6
**Confidence:** 4

**Summary:**

This paper propose a method to generate unlearnable examples for pretrained models. The pretrained model can be robust to unlearnable examples generated by previous methods since there is prior knowledge and model is able to ignore the shortcut. This paper propose to change the learning target while optimizing the perturbation, which violates the model prior and thus generates effective perturbation. The experiments are conducted on different datasets and model backbones and show consistent improvement on misguiding the model learning proess. Compared with other methods targeting for model-from-scratch, this method works better on pretrained models.

**Strengths:**

The proposed strategy for generating perturbations with changed target effectively attack the learning process of a pretrained model.

Some analysis on parameter updates provide insights on difference between scratched model and pretrained model.

Adequate ablations on datasets and model backbones to show the generalizability.

The comparison is made on both randomly initialized and pretrained surrogate model, both showing good improvement with previous methods

**Weaknesses:**

No major weaknesses in this paper. See the questions part for minors.

**Questions:**

In figure 3, what samples are used to test the surrogate model accuracy? Are they used in the perturbation optimization step?
Also, just to confirm, is the perturbed data in figure 3 generated with the optimized surrogate model or it is changing as the perturbation optimization goes on?

In table 1, other methods are under the setting where the surrogate model is randomly initialized, what is the result if testing the proposed method under this setting? This could be a strong proof of generalizability

---

> ### Author Response · Authors · 2025-11-24
> **Response to Reviewer TKJA**
>
> We thank reviewer TKJA for the constructive feedback and valuable suggestions. Below, we provide responses to each raised question. Please feel free to reach out if any further clarification is required.
>
> ## **Q1:** Clarification on the samples used to test the surrogate model in Figure 3.
>
> We clarify that when evaluating training accuracy on CIFAR-10 in Figure 3, we are using two variants of CIFAR-10 training samples to test the surrogate model, namely the **original clean** training samples and the **perturbed** training samples. The **huge performance gap** on these two samples reflects that perturbations gradually **mislead** the real data-label association within pretraining priors and force the surrogate model to rely on **spurious correlations** between perturbations and labels for prediction. Therefore, the surrogate model fails to obtain satisfactory results on clean samples. We also clarify that the perturbed data are **changing** as the added perturbations are continually optimized through our BAIT framework. We have added corresponding discussions in **Section 4.4**.
>
> ## **Q2:** Test our method with a randomly initialized surrogate model.
> We value your advice and optimize our method with a **randomly initialized** surrogate model. We then evaluate its performance against ImageNet-pretraining priors on three distinct datasets and add results in **Appendix F.1**. As shown in **Table R1-1**, our method outperforms baseline methods and exhibits strong capabilities in rendering data unlearnable, which further demonstrates its effectiveness against pretraining priors.
>
> **Table R1-1.** Test accuracy (%) ↓ comparison against ImageNet-pretrained models with a randomly initialized surrogate model.
> |  Dataset  |  EMN  |  TUE  |  REM  |  LSP  |  GUE  |  14A  | Ours (with randomly initialized surrogate) |
> | :-------: | :---: | :---: | :---: | :---: | :---: | :---: | :----------------------------------------: |
> | CIFAR-10  | 61.82 | 82.72 | 74.46 | 54.20 | 23.17 | 65.70 |                 **15.97**                  |
> | CIFAR-100 | 55.53 | 55.20 | 55.22 | 36.98 | 54.09 | 33.67 |                 **22.83**                  |
> |   SVHN    | 37.55 | 41.15 | 76.45 | 38.91 | 39.68 | 81.47 |                 **17.93**                  |

---

### Author Response · Authors · 2025-12-03
**Summary Comments**

Dear AC/SAC/PC,

We sincerely thank the reviewers, AC and PC for their efforts during the review process. Given the absence of reviewer engagement during the earlier discussion period, we would like to provide a final summary of our contributions, the reviews, and the responses to facilitate your final assessment.

## **Our Core Contributions**
In this work, we present the **first** systematic investigation exposing the **fundamental vulnerability** of UEs when confronted with **pretrained models**. To address this limitation, we introduce BAIT, a pioneering framework that actively **neutralizes pretraining priors** to generate **robust UEs** effective against pretrained models.

We are encouraged that the reviewers recognized the following strengths:
- **Motivation:** Reviewers found our empirical analysis regarding the discrepancy between UEs applied to train-from-scratch versus pretrained models to be insightful (`Reviewer TKJA`), our identified vulnerability of UEs against pretraining priors to be practically realistic (`Reviewer vgQM`).
- **Method Design:** Reviewers stated BAIT as an effective (`Reviewer TKJA`), pioneering, and convincing (`Reviewer rRcL`) approach.
- **Experimental Strength:** Reviewers found our comparisons and ablations to be adequate and the generalizability to be strong (`Reviewer TKJA`), our UEs to be effective in substantially reducing test accuracy (`Reviewer TKJA`, `vgQM`, `rRcL`), and our comparison to be fair (`Reviewer vgQM`).
- **Writing:** Reviewer found our writing and presentation to be of high quality (`Reviewer rRcL`).

## **Clarification on Major Misunderstandings**
We observe that the negative assessments of `Reviewers vgQM` and `rRcL` stem from specific misunderstandings regarding our novelty and method design. We have provided comprehensive responses to clarify these major points, which we believe clearly highlight our contribution and significance to the UE community. Regrettably, neither reviewer engaged further during the discussion period.

- **BAIT $\neq$ Error-Minimizing Noise.** We believe that the negative assessment of `Reviewer vgQM` came from the **conflation** of BAIT and error-minimizing noise. We emphasize that BAIT is **fundamentally different**. As detailed in our specific response (*Response to Reviewer vgQM 1/3, W1*), we have strictly delineated these distinctions across three key dimensions: **(1) Data-Label Association** (core rationale), **(2) Formulation** (mathematical modeling), and **(3) Optimization Strategy** (algorithmic solution).
- **Pioneering Investigation into the Vulnerability to Pretraining Priors.** Regarding `Reviewer rRcL`, we believe the major negative assessment is based on **some misunderstandings on the UE literature review**, where the reviewer stated that “*this issue has been expressed in many previous studies*” but did **not provide corresponding references**. While the reviewer did mention two papers in the end, upon a detailed review of the referenced papers and the broader UE literature, we confirm that the focus of these papers is **different** from ours and the vulnerability of UEs to pretraining priors **has not yet been investigated**. To the best of our knowledge, our work is the **first systematic attempt** to **explicitly identify** and **address** this limitation. Please refer to *Response to Reviewer rRcL (1/5) W1* for detailed response.

## **Summary of Scores**
- `Reviewer TKJA`: 6 – Positive. Stated there are “***no major weakness in this paper***” and noted the additional verification with a randomly initialized surrogate “***could be a strong proof of generalizability***”.
- `Reviewer vgQM`: 4 – No response received during the discussion period.
- `Reviewer rRcL`: 4 – No response received during the discussion period.

## **Summary of Revision to Address the Raised Concerns**
- For `Reviewer TKJA`: We provided the requested implementation details for Figure 3 and added verification with a randomly initialized surrogate model.
- For `Reviewer vgQM`: We clarified the fundamental distinctions between BAIT and error-minimizing noise. Furthermore, we provided details on **perturbation optimization**, the **fine-tuning setup**, and **surrogate usage**, alongside additional evaluations against **advanced defenses**.
- For `Reviewer rRcL`: We included additional **evaluations** on **larger models** and **datasets**. We also added **analyses** on the dynamics between pretraining and fine-tuning, the **relationship** between parameter updates and performance, **transferability** to different downstream tasks, and further **verification** of the data protection ratio.

Regrettably, reviewers did **not** engage during the discussion phase. We entrust the AC with making the final, expert determination based on the presented clarifications and results. We hope this summary assists with your final assessment. Thank you once again for your time and consideration.

Best regards,\
Submission 4585 authors

---

### Meta-Review · Area_Chair_cCq5 · 2026-01-09

**Summary:**

This paper investigates unlearnable examples (UEs) in the setting of fine-tuning from pretrained models and proposes BAIT, an optimization-based approach based on pretraining-induced label associations. The reviewers generally acknowledged that the problem setting is realistic, the proposed method is technically sound, the experimental section is extensive, and the empirical results demonstrate strong attack effectiveness in fine-tuning scenarios.

The reviewers raised several concerns and questions, particularly regarding the novelty against [1], robustness against realistic defenses such as JPEG compression and orthogonal projection, generalizability on large-scale models and larger datasets, transferability to other downstream tasks, and verification of data poisoning ratios.

In response, the authors provided detailed description of bi-level optimization, new defense evaluations (JPEG compression and orthogonal projection), experiments on larger pretrained ViT models and larger datasets (Flowers and ImageNet subsets), and explicit analyses showing that performance drops on the original ImageNet data during fine-tuning are comparable for clean and poisoned data, indicating that BAIT does not simply induce excessive forgetting.

While the novelty and mechanistic interpretation could be sharpened, the paper addresses an important gap in the UE literature and provides a technically solid and empirically well-supported solution. AC therefore recommends acceptance.

[1]  Unlearnable Examples: Making Personal Data Unexploitable. 2021

**Reviewer Concerns:**

addressed: TKJA, rRcL
partially addressed: vgQM

**Reviewer Scores:**

TKJA: 6->6
vgQM: 4->4 or 4->6
rRcL: 4->6

---

### Decision · Program_Chairs · 2026-01-26

Accept (Poster)